# Direct detection of a single photon by humans

Jonathan N. Tinsley[1,2,†,*], Maxim I. Molodtsov[1,2,3,*], Robert Prevedel[1,2,3], David Wartmann[1,†], Jofre Espigulé-Pons[2,4], Mattias Lauwers[1] & Alipasha Vaziri[1,2,3,5]

Despite investigations for over 70 years, the absolute limits of human vision have remained unclear. Rod cells respond to individual photons, yet whether a single-photon incident on the eye can be perceived by a human subject has remained a fundamental open question. Here we report that humans can detect a single-photon incident on the cornea with a probability significantly above chance. This was achieved by implementing a combination of a psychophysics procedure with a quantum light source that can generate single-photon states of light. We further discover that the probability of reporting a single photon is modulated by the presence of an earlier photon, suggesting a priming process that temporarily enhances the effective gain of the visual system on the timescale of seconds.

[1] Research Institute of Molecular Pathology, Dr Bohr-Gasse 7, 1030 Vienna, Austria. [2] Research Platform Quantum Phenomena & Nanoscale Biological Systems (QuNaBioS), University of Vienna, Dr Bohr-Gasse 7, 1030 Vienna, Austria. [3] Max F. Perutz Laboratories, University of Vienna, Dr Bohr-Gasse 9, 1030 Vienna, Austria. [4] Faculty of Physics, VCQ, University of Vienna, Boltzmanngasse 5, 1090 Vienna Austria. [5] Laboratory of Neurotechnology and Biophysics, The Rockefeller University, 1230 York Avenue, New York, New York 10065, USA. * These authors contributed equally to this work. † Present address: Department of Physics, University of Liverpool, Liverpool L69 7ZE, UK (J.N.T); Department of Chemistry, University of California, Berkeley, 313 Lewis Hall, Berkeley, California 94720, USA (D.W). Correspondence and requests for materials should be addressed to A.V. (email: vaziri@rockefeller.edu).

Landmark experiments by Hecht and colleagues in the 1940s established that dark-adapted human subjects are capable of reporting light signals as low as a few photons ($\sim 5$–$7$)[1]. However, whether evolutionary pressure has pushed the visual system and the post-processing performed by the retina and brain to detect a single photon has remained an open question[1–6]. The answer to this question is of fundamental significance as it could provide insights into the mechanisms underlying the limits of evolutionary optimization, as well as open up fundamentally new avenues for probing retina-signalling pathways using quantum states of light[7–9]. Moreover, as noise is omnipresent at all stages of the visual system[4,10–12], it has been suggested that using light with lower statistical variability than that of classical Poissonian light might help to directly extract characteristics of intrinsic noise that limits detection at low-light levels by removing the input noise[4,13]. Thus, understanding the mechanism underpinning the detection of a single photon with only $\sim 4 \times 10^{-19}$ J of energy would demonstrate that the weakest possible, quantized optical signal is not completely swamped by neuronal noise and other inefficiencies. This may help to uncover more generally how biological signal detection at the absolute physical limits is implemented.

Previous experimental attempts to approach these questions were hampered by the lack of appropriate technologies, low-experimental statistics in the lowest-photon range, and non-ideal psychophysical procedures[2,3,14,15]. Most importantly, all experiments so far have been performed with Poissonian light sources such as attenuated laser pulses, which exhibit an intrinsic and irreducible variability in the actual number of photons emitted. Together, these limitations have led to an inherent ambiguity about the exact number of photons required to elicit the perception of seeing light[4]. By addressing the above shortcomings, we demonstrate that humans can detect a single-photon incident on their eye with a probability significantly above chance. Additionally, our results lead us to hypothesize that under such extreme light conditions, the absorption of a photon induces a modulation of the gain in the visual system with a characteristic temporal evolution persisting on the order of seconds.

## Results

**Design of quantum optical single-photon light source.** To probe the absolute limit of light perception, we built a single-photon quantum light source with sub-Poissonian photon number statistics based on spontaneous parametric down-conversion (SPDC). SPDC is a quantum optical technique in which correlated pairs of photons (called signal and idler) are produced probabilistically from a higher energetic pump photon in a non-linear crystal following energy and momentum conservation[16,17] (Fig. 1a). By detecting one of the photons (idler) and sending the other (signal) to the observer's eye, our SPDC source allowed us to create an effective single-photon light source with a sub-Poissonian photon number distribution (Fig. 1a, Supplementary Fig. 1 and see Methods section for details). Ideally, such a source features a small ratio of multiple-to-single photon emissions. To optimize this ratio in practice, we used the multi-pixel sensor of an electron multiplying charge-coupled device (EMCCD) camera as our idler detector (Fig. 1a). As the EMCCD can detect multiple photons simultaneously, it allowed us to identify and reject, that is, post-select, all events other than those where a single-photon pair was generated with a higher efficiency than with more traditional single-photon avalanche diodes (SPAD) (Fig. 1b and Methods section).

In general, lower SPDC pump powers are beneficial, as they generate fewer multi-photon events and thus provide a lower ratio of multiple-to-single photons (Fig. 1b). However, this in turn significantly increases the number of experimental trials required to demonstrate single-photon perception above the chance level, as a lower pump power leads also to a higher number of blank events (Methods section). We performed numerical simulations to quantify this trade-off (Fig. 1c) and found an optimal choice of mean number of produced photon pairs per pump pulse (0.048) that still allowed for a feasible number of experimental trials. Events in which single photons were sent into the eye (single-photon events) were identified after the experiment and used for further analysis by post-selecting the cases where one and only one detection was registered by the EMCCD in the idler arm. Under these chosen experimental conditions, multiple-photon pairs were produced only in $\sim 0.11\%$ of all trials. The two-dimensional detector architecture of the EMCCD allowed identifying the majority of such two- and multi-pair events ($\sim 80\%$), such that in post-selected trials the probability of having two or more photons producing the retinal signal was only $\sim 0.02\%$ (Methods section). This means that from all our trials that passed post-selection on average $< 1$ would have led to a two or multi-photon retinal signals. Thus, unlike in previous studies, we can exclude the possibility that the subjects' responses in our experiments are due to such two- or multiple-photon events.

**Humans can detect a single-photon incident on the eye.** Subjects were presented with light stimuli from the SPDC source while using a modified version (Methods section) of the two-alternative forced-choice (2AFC) protocol[18,19]. In our case, in each trial the subject had to identify a light stimulus from a blank delivered in a temporally separated fashion. After subjects provided their response, they received feedback as to whether their response was correct. Given that only in a small fraction of cases an actual photon pair was generated by the SPDC source, this protocol allowed us to generate a large number of catch trials (Methods section). In addition, after the subject had provided a response they were asked to rate their confidence in the response on a trichotomous scale, R1–R3 (Fig. 1a).

Averaging across subjects' responses and ratings from a total of 30,767 trials, 2,420 single-photon events passed post-selection and we found the averaged probability of correct response to be $0.516 \pm 0.010$ ($P = 0.0545$; Fig. 2a), suggesting that subjects could detect a single photon with a probability above chance. This conclusion was further corroborated by additional experiments based on an attenuated Poissonian light with a mean photon number of one. Given that for such a source the probability that two or more photons lead to light induced, multiple-photon events at the retina is only $\sim 3.7\%$ allowed us to use both data sets to test the same hypothesis and obtain a more significant $P$ value of 0.014 using Fisher's method (Supplementary Peer Review File, Fig. 1).

Next we investigated the distribution of subjects' confidence ratings for our single-photon SPDC source. As expected, given the weak stimulus, the distribution of confidence ratings for correct responses was dominated (88%) by low confidence R1 and R2 responses (Fig. 2b). Considering only the answers with the high-confidence R3 rating, we found that the probability of providing the correct response was significantly elevated compared with all responses ($0.60 \pm 0.03$, $P = 0.0010$), which demonstrates that subjects indeed detected a single photon in the high-confidence trials (Fig. 2a).

Not every single photon incident on the eye leads to an isomerization and a subsequent production of a retinal signal. Based on the efficiency of the signal arm and the visual system, we estimate that in $\sim 6\%$ of all post-selected events an actual light-induced signal was generated (Methods section). Therefore, it is expected that from all trials only this fraction should be able

to contribute to an above chance performance as well as to some increase in the subjects' choice of high-confidence (R3) ratings. Thus, the correlation between the statistically significant sensitivity of subjects for a single-photon stimulus with their higher confidence rating provided us with further corroborative evidence that subjects could indeed detect a single photon. To convince ourselves further that the observed performance of subjects would be within a plausible regime, we used signal detection theory to compare our subjects' performance with the expected performance of an ideal detector, whose operation was only limited by intrinsic noise and efficiency (Supplementary Note 1, Supplementary Fig. 2). We found that the performance of our subjects did not exceed the performance of an ideal detector

(Fig. 2a) for a plausible range of reported noise and efficiencies in the literature (Supplementary Table 1).

Finally, we systematically excluded a set of alternative explanations as the basis for our observations, such as subjects' bias (Supplementary Note 1) or possible contamination with background light (Methods section). We also confirmed the overall performance of our subjects and the experimental set-up including our psychophysics protocol by reproducing the results of previous experiments in the higher photon range using a classical Poissonian light source (Supplementary Fig. 3)[2,3,14]. We found the overall shape of the psychometric curves for all subjects, its characteristic parameters such as the threshold and the index of discriminability $\Delta m$, a measure of stimulus discrimination sensitivity[19] (Supplementary Note 2), to be consistent with previous observations[2,3,14] (Supplementary Fig. 3).

**Single-photon perception is enhanced by an earlier photon.** Led by the observation that single photons could be detected by subjects, we asked whether perception at these extreme light conditions is limited only by dark isomerization noise events. We analysed how the probability of correct response in single-photon post-selected events depends on the time to the previously registered photon in the idler arm, irrespective of the number of trials separating the two events. Surprisingly, a strong dependence on the temporal separation of the two events was observed peaking at $\sim 3.5$ s, with a decay time on the order of seconds (Fig. 2c). Such a long timescale phenomenon represents more than an order of magnitude disparity with the known integration time of the visual system[4]. This result directly shows that the probability of correctly reporting a single photon is highly enhanced by the presence of an earlier photon within $\sim 5$ s time interval. Averaging across all trials that had a preceding detection

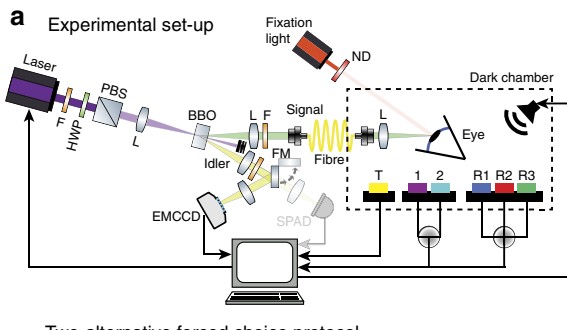

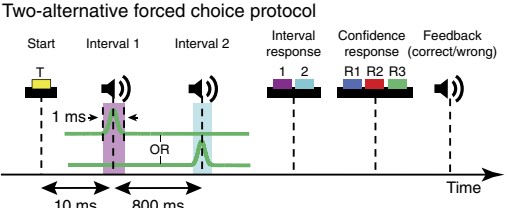

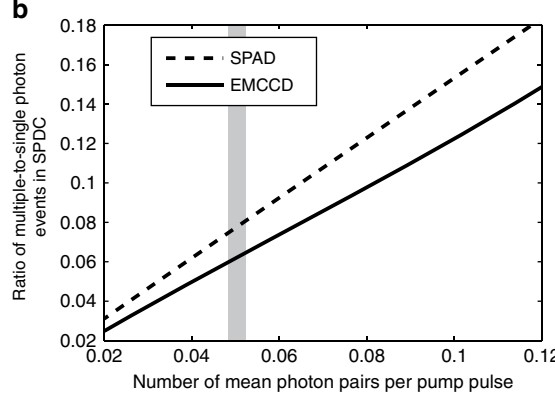

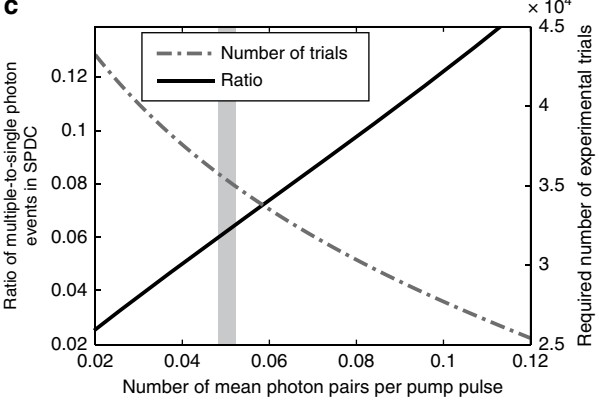

**Figure 1 | Schematics of the experimental set-up and characterization of the single photon quantum-optical light source.** (**a**) Top: schematic of the set-up for generating single photons via SPDC. Light stimuli are triggered (T) by subjects and coupled into a single-mode fibre entering the dark chamber. The light is directed and focused onto the pupil (Maxwellian view) at an angle of 23° temporal to a highly attenuated red fixation light presented at the fovea. BBO, beta-barium borate crystal; F, bandpass and spatial filter; HWP, half-wave-plate; L, lens; ND, neutral density filter; PBS, polarizing beamsplitter. A flip mirror (FM) allowed for the switching of the detection mechanism from the EMCCD to a SPAD (semi-transparent, see Methods section for details). Bottom: schematic of the 2AFC protocol. A 1 ms light pulse is presented together with either the first or the second acoustic signal (intervals 1 and 2) that are separated by 800 ms. After the second acoustic signal, the subject gives their answer (1, 2) and a confidence rating (R1, R2 and R3; see Methods section for details.) (**b**) Numerical simulation of multiple-to-single photon ratio in SPDC as a function of mean photon pairs per pulse (that is, at different pump laser intensities). While the multiple-to-single photon ratio decreases as a function of mean photon pairs for both detection schemes, the EMCCD (solid line) exhibits an improved multiple-to-single ratio compared with the SPAD (dotted line), as the multi-pixel based detection in case of EMCCD allows for identification and of two and multi-photon events and their rejection. (**c**) Trade-off between multiple-to-single photon ratio and required number of experimental trials. Multiple-to-single photon ratio in SPDC using EMCCD-based detection as a function of mean photon pairs per pulse (black solid line – same as in **b**) exhibits an inverse relationship with the number of trials required to discriminate the performance of an ideal detector from random chance level (0.5) with a statistically significant probability (95% confidence interval; grey dotted line). The vertical grey bar indicates the mean number of photon pairs per pump pulse that was used in the experiments (see Methods section for details).

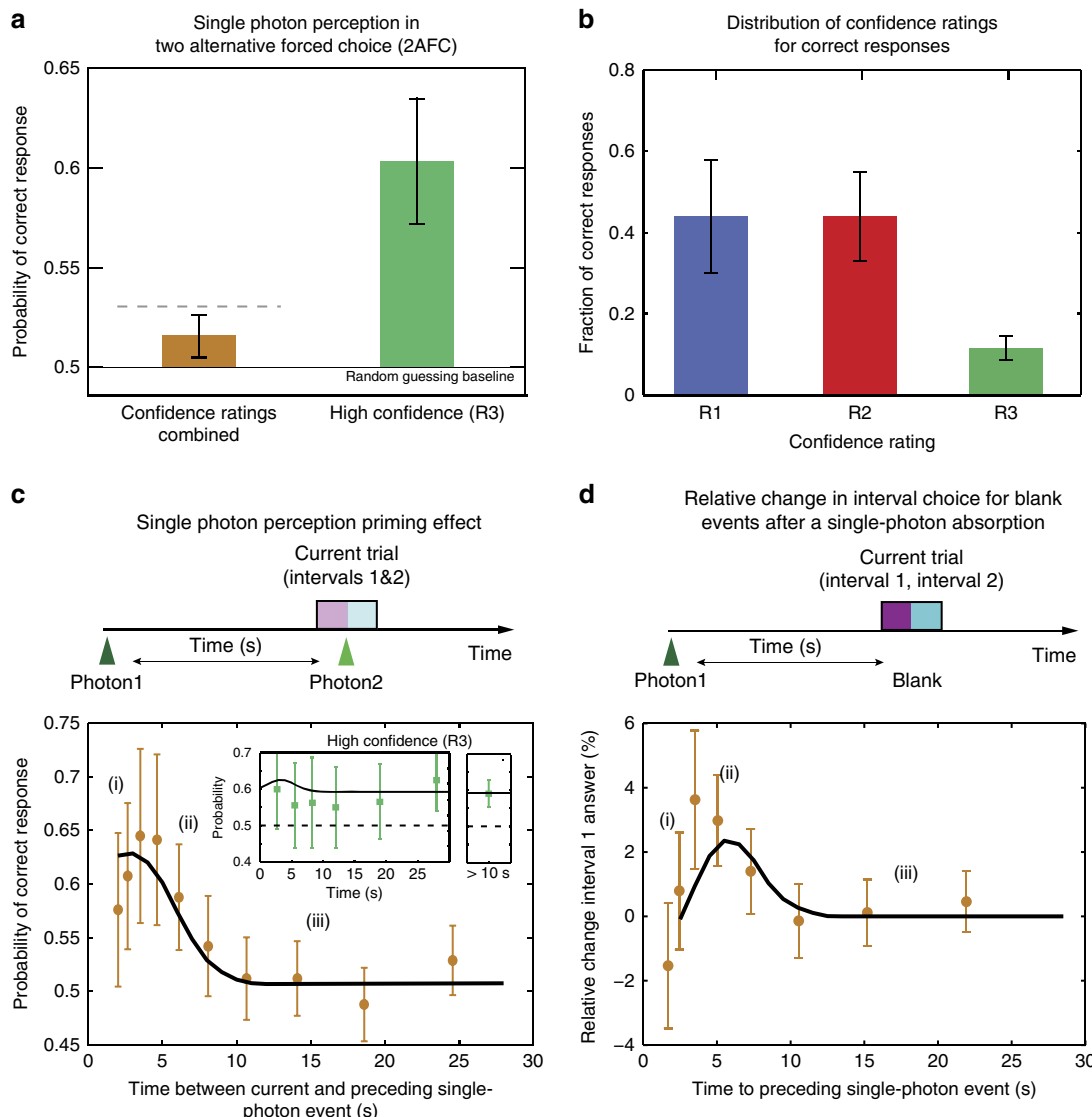

**Figure 2 | Single-photon perception and photon-induced temporal modulation of detection probability. (a)** Perception of a single photon. Probability of providing the correct response in 2AFC trials for all post-selected single-photon events (brown, $n = 2,420$) and high-confidence R3 responses (green, $n = 242$) averaged across all subjects. 0.5 is the baseline and corresponds to random guessing. The horizontal dashed line indicates the upper theoretical limit of performance of an ideal detector operating at the physiological level of detection efficiency in the absence of additional noise and using the framework of signal detection theory (Supplementary Note 1). **(b)** Distribution of different confidence ratings for post-selected single-photon events in which the stimulus was correctly identified. **(c)** Probability of correct response as a function of the time to the preceding single-photon event, data averaged across subjects and ratings ($n = 2,420$). The probability of correct response for events combined at times between 0 and 10 s is significantly higher than for events at longer times ($>10$ s, $P = 0.02$). 0.5 is the baseline and corresponds to random guessing. (i–iii): three temporal regions corresponding to different scenarios of the model in Fig. 3a. Solid line is a fit to the model illustrated in Fig. 3a and discussed in Supplementary Note 3. Inset shows the same data but for the high-confidence rating only ($n = 242$). The additional panel on the right is the probability of correct response for all combined R3 events outside of the temporally enhanced detectability region ($>10$ s). **(d)** Relative change in interval 1 responses for blank trials, that is, containing no photons ($n = 28,233$) as a function of time to the preceding single-photon event. The data are averaged across subjects and ratings. (i–iii): Three temporal regions corresponding to different scenarios in Fig. 3b. Combined data in the range 3–6 s is significantly above 0% ($P = 0.007$). The solid line is a fit to the model illustrated in Fig. 3b ($\chi^2 = 2.7$) and discussed in Supplementary Note 3. In **a–d** error bars denote s.e.m.

within a 10-s time window, the probability of correct response was found to be $0.56 \pm 0.03$ ($P = 0.02$). Outside of this temporary enhanced region ($>10$ s), the probability of correct response was not significantly above chance level ($0.510 \pm 0.011$, $P = 0.2$) when averaged across all rating responses, but it was significant ($0.59 \pm 0.04$, $P = 0.02$), for high-confidence R3 responses (inset Fig. 2c). These results demonstrate that in high-confidence R3 trials subjects can detect a single-photon irrespective of the presence of a single photon in the preceding trial. This means that

while the discussed photon-induced enhancement of the gain leads to a transient enhancement of the probability of detection, it is not strictly required for single-photon perception.

To more directly characterize the signature of such a transient modulation of the visual system, we analysed the trials failing the post-selection, the vast majority of which did not contain any stimulus photons. For these trials, we observed an increased probability for choosing the first-time interval for time delays up to $\sim 4$ s following the detection of the preceding photon

(Fig. 2d), with a decay on the same timescale as in Fig. 2c. This demonstrates that the absorption of a single photon in the visual system produces persisting and significant behavioural differences that could be detected through our 2AFC paradigm. Thus, consistent with both observations, we suggest that the detection of a single photon – or equally a photon-like noise event (that is, spontaneous isomerization) – temporally increases the effective gain of the visual system under extreme low-light conditions, such that a second temporally coinciding photon (or photon-like noise event) can be behaviourally detected with a higher probability.

We aimed at finding a model that would explain our above observations (Fig. 2c,d) while at the same time making a quantitative prediction about the expected distribution of confidence ratings. To do so we devised a model in which the probability of detection by the subject was proportional to the product of the combined number of photons and photon-like noise events with the system's gain (Fig. 3a,b). The gain's photon-induced characteristic temporal profile (Fig. 3c) was extracted by fitting the data (Fig. 2c,d) to the model (Fig. 3a,b, Supplementary Note 3). Within this model the higher probability of correct response for short delays (Fig. 2c) can be understood as

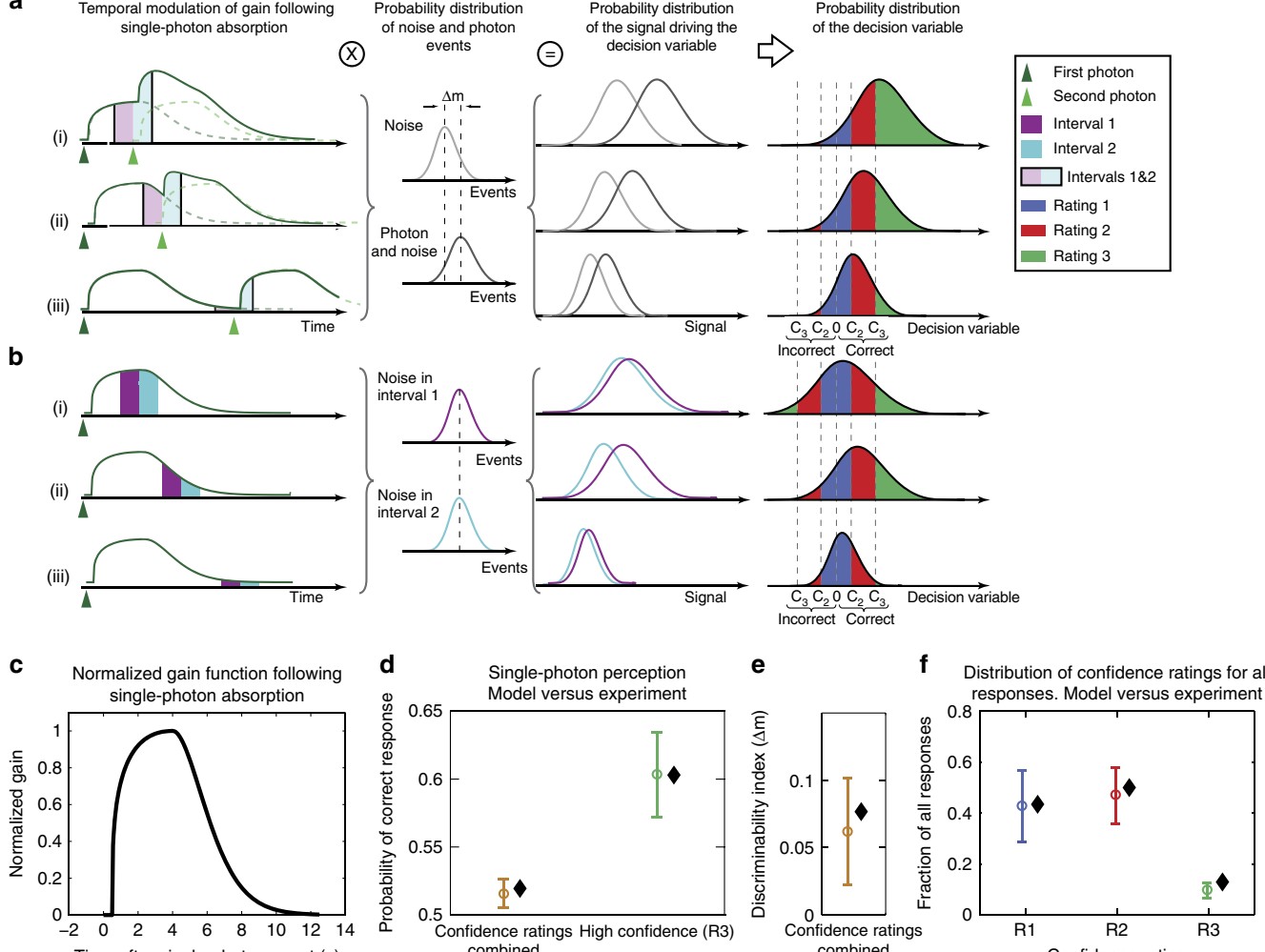

**Figure 3 | Model for photon-induced priming of single-photon detection probability.** (**a,b**) First column: temporal modulation of the effective gain following the absorption of two (**a**) or one (**b**) photons separated by three different time delays, corresponding to the respectively regions denoted in Fig. 2c,d (i–iii). Second column: probability distribution for photon plus photon-like noise events and photon-like noise events only during intervals 1 and 2. $\Delta m$ indicates the discriminability index, a measure for the separation between the noise and the signal plus noise distributions (see Supplementary Note 2 for details). Third column: accumulated signal driving the decision variable, that is, the product of the gain (left) and the probability distribution of the photon and photon-like noise events (**a**) or noise events only (**b**) for the two intervals of a trial. Fourth column: the decision variable, which is proportional to the difference between the signal distributions. The decision criteria $C2$ and $C3$ separate the decision variable into low, medium and high-confidence ratings (Supplementary Note 3). Rows (i–iii) refer to the three temporal regions shown in Fig. 2c. The distributions shown are cartoons. For the actual parameter values refer to Supplementary Note 3. (**c**) Normalized temporal profile of the gain following a photon absorption at $t = 0$. Kernel obtained by simultaneously fitting data in Fig. 2c,d to the model (Supplementary Note 3). (**d**) Probability of correct response in all single-photon experiments (circles with error bars – same data as in Fig. 2a) and fit to the model including decision criteria for high-confidence R3 (black diamonds). 0.5 is the baseline in 2AFC and corresponds to random guessing. Experimental data and fits are separated horizontally for clarity. (**e**) Discriminability index ($\Delta m$; see Supplementary Note 2) based on all single-photon experiments (circle with error bar) and fit to the model (black diamonds). (**f**) Distribution of all answers for post-selected single-photon events for different confidence ratings (same data as in Fig. 2b, circles with error bars) and the fit to the model including decision criteria (black diamonds). Goodness of the fit for the data combined in **d,f** is $\chi^2 = 0.012$.

a result of an increase in the system's gain due to the previously absorbed photon, irrespective of the 2AFC time interval in which the photon was presented (Fig. 3a). The relative change in the fraction of interval 1 answers (Fig. 2d) is due to the temporal separation of trials resulting in a higher contribution of the gain to the amplification of a photon-like noise event in the first compared with the second time interval (Fig. 3b). Furthermore, using the extracted gain kernel (Fig. 3c), the actual delays between individual single-photon events in the experiment and by introducing decision criteria used by the subjects to assign their confidence ratings (Fig. 3a,b, Supplementary Note 3), our model could quantitatively explain the experimentally observed distribution of ratings and the probability of correct response for each rating in our single-photon experiments (Fig. 3d–f; Supplementary Note 3).

## Discussion

To our knowledge, these experiments provide the first evidence for the direct perception of a single photon by humans. Previous psychophysics studies have been performed only in a regime where multiple photons could reach the retina due to both the used light intensities and the technological lack of a light source with a sub-Poisson photon number distribution. The explanation of the corresponding psychophysics data required either assuming unrealistically low-quantum efficiencies or the assumption of an additional multiplicative noise[1–3]. However, our single-photon data show that the subjects' performance (Fig. 3d–f) can be readily accounted for by physiological values of the overall quantum efficiency of the eye (Supplementary Table 1) and a Poisson distributed dark isomerization rate suggesting single-photon perception is not limited by further down-stream inefficiencies of the visual system and the brain.

Our light source and experimental protocol allowed us to further discover a single-photon-induced priming effect that is characterized by a temporal modulation of the gain of the visual system. It is well established that the extreme dynamic range (approximately nine orders of magnitude)[20] and sensitivity of the visual system is mediated by a light-dependent modulation of the system's gain. Although over a wide range the sensitivity of the visual system is inversely proportional to the average light level, our data show that at the single-photon level, a higher transient detection probability for a photon is obtained if another photon was absorbed previously. This phenomenon is conceptually reminiscent of coincidence detection, although at timescales orders of magnitude slower and likely involving a different physiological mechanism from the known retinal circuit computations[21,22].

The retinal and brain-circuitry mechanisms underlying our above observations remain the scope of future work. Further studies aimed at different scales and combining different methodologies such as electroencephalography[15] or magnetoencephalography[23] with psychophysics might allow to elucidate possible role of top–down cortical effects, including visual attention or brain oscillations, in the subjects' performance and choice of confidence ratings. In addition, post-mortem physiological studies on the human retina would be likely required to elucidate any involved retinal circuits mechanisms on the cellular level.

Furthermore, given the inherent quantum nature of light and the fact that single photons can be prepared in superposition states of space, time, energy and other degrees of freedom, our study opens up new, speculative possibilities to investigate to what extent such states may also bear unique physiological signatures as well as to test theoretical proposals that use human observers for experimental tests of quantum non-locality[24–27]. Finally, using quantum technologies to generate sub-Poissonian photon number distributions allows experiments to be performed

with a reduced input noise into the visual system and consequently facilitates a more direct access to the internal system's noise, which may provide new avenues for probing yet undiscovered retinal pathways.

## Methods

**Psychophysics experiments protocol.** Three subjects participated in the experiments. All were male, in their twenties and had optimal vision, in Subject A's case corrected by contact lenses. The experiments were approved by the ethics committee of the University of Vienna and participants participated entirely voluntarily and were fully informed of the aims and methods of the experiments.

All experiments were conducted inside a light-proof chamber ($\sim 1 \times 1 \times 2$ m) located in a dark room. The subjects wore headphones through which they could hear sounds that heralded the light and control pulses, respectively, as well as feedback on their response. Foam-filled panels provided good acoustic isolation such that no visual or auditory cues were available that could be potentially used by the subjects to infer the time interval in which the single photon stimulus was presented. After a period of $\sim 35$–40 min, a dark-adapted subject fixated with his right eye on a barely visible red light, presented normal to the cornea, with the subject's head kept in position by a bite bar and a headrest. The fixation light was a 660 nm LED (full-width at half-maximum = 5 nm) operated at $\sim 6\,\mu$W output power, which was further attenuated with OD 6.5 filters before impinging normal to the cornea. The stimuli were presented in a Maxwellian view (that is, light focused onto the pupil) $\sim 23°$ temporal to the fixation light and the main optical axis of the eye. The chosen position ensured that the light stimuli were presented to the location on the retina with the highest density of rod cells, the main mediators of scotopic vision.

The subjects triggered trials by themselves and were instructed to proceed at whatever pace they felt comfortable. On the triggering of a trial, the subject was presented with two intervals each of 1 ms duration and separated by $\sim 800$ ms, with the initiation of each interval being heralded by a synchronous acoustic signal of $\sim 10$ ms duration. One trial on average took $\sim 2.5$ s, but subjects could pause for a longer time if they wished to do so. One of the two intervals was pseudorandomly chosen to trigger a possible single photon from SPDC around the peak rod sensitivity ($\sim 500$ nm), while the other interval was a 'blank'. After the second interval the subject indicated which of the two intervals they thought contained the stimulus photons (interval response) and thereafter provided a confidence rating (R1–R3; Fig. 1a). Subsequently, subjects received acoustic feedback as to whether their interval choice matched the randomly chosen interval to contain the stimulus. This feedback helped subjects to remain alert and motivated throughout the entire session. The prerequisite is, however, to ensure that subjects did not use any cues that allowed them to infer the time interval into which the photon was emitted. To do so we analysed the performance of subjects in single-photon SPDC trials where no photons were detected by the EMCCD and could use these as catch trials. In the majority of these trials ($\sim 95\%$) no photon pair was generated and thus no photons were actually sent to the subjects' eye. We found that in these catch trials the probability of correct response was not different from the 0.5 baseline for both, the combined and the high-confidence responses ($0.505 \pm 0.003$, $P = 0.08$ and $0.507 \pm 0.01$, $P = 0.3$, respectively).

Before collecting data, subjects were extensively trained using a classical light source with photon number between 1 and 15 photons at the cornea (Supplementary Fig. 4a). The improved performance with experience is clearly and quantitatively visible (Supplementary Fig. 4b,c). Subjects typically required 6–8 sessions, performing one session a day, to reach their optimal performance level (Supplementary Fig. 4b,c). Each session took $\sim 2$ h, when including dark adaption.

During data acquisition, each subject went through up to 20 sessions. Still this high amount of sessions was not enough to obtain statistically significant performance for individual subjects, and therefore we pooled the data together to increase significance (Fig. 2a–d). As subject's sensitivity and criteria used to assign the confidence ratings might vary in psychophysics trials[28], we aimed to minimize or normalize possible factors causing variability to achieve maximum sensitivity and similarity across subjects by using extensive training of the subjects and using our 2AFC paradigm.

Since subjects showed significantly higher performance for the high-confidence rating R3 events (Fig. 2a), we analysed the statistical distribution of these events in more detail. The time between correct high-confidence responses was well described by an exponential distribution ($R^2 = 0.98$) suggesting these correct events occur randomly in time, as expected (Supplementary Fig. 5a). In addition, the distribution of the times when correct high-confidence responses occurred during a session (Supplementary Fig. 5b) was not significantly different from a uniform distribution ($P = 0.07$, Kolmogorov–Smirnov test) demonstrating that the performance of the subjects is approximately constant and that subjects performed equally well throughout the whole duration of the session.

Finally, we verified that performance in single-photon trials was not correlated with the feedback received in the previous trial, suggesting that the observed effect was not due to increased attention on the part of the subject (Supplementary Fig. 6).

**SPDC set-up and components.** In SPDC high-energy photons decay spontaneously in a non-linear crystal without inversion symmetry under energy and momentum conservation into two lower energetic photons called the signal

and the idler. Typically only $\sim 10^{-9}$–$10^{-12}$ of the incident photons lead to the generation of a pair, but due to conservation of energy and momentum, the photons of a pair are always generated together and in a correlated manner.

A diagram of the experimental set-up for the SPDC source experiments is shown in Fig. 1a. To prevent background light from impacting upon the experiment, the whole-optical SPDC set-up was constructed within a light-proof container. The single photons are produced via SPDC inside a 1-mm-thick beta-barium borate crystal cut for non-degenerate type-I phase-matching at the desired wavelengths ($\sim 504$ and $561$ nm; $47.6^\circ$). The crystal was pumped with 10 ns, 266 nm laser pulses from a diode-pumped, passively Q-switched laser, capable of triggered operation of up to 10 kHz. We chose this non-degenerate configuration such that the wavelength of idler and signal mode coincided with the maximum quantum efficiency of our custom-coated Andor iXon Ultra EMCCD camera and the peak of the human rod response, respectively. The frequency modes were selected by filters (Semrock LL02-561-12.5; $\lambda = 561.4 \pm 1.1$ nm and Thorlabs FELH0500 + Semrock TBP01-501/15, respectively) and translatable irises. A flip mirror in the idler path allowed for the switching between the EMCCD and SPAD detectors. Both the EMCCD and the SPAD were used to measure $g^{(2)}(0)$ values (Supplementary Fig. 1), while only the EMCCD was used in single-photon experiments with actual subjects. Both detectors were never used simultaneously.

The signal mode was coupled into a single-mode fibre with $\sim 40\%$ efficiency after spectral filtering. The angle of the tunable TBP01-501/15 filter relative to the optical path allowed for the fine-tuning of the bandwidth to $\sim 500$–$508$ nm, with absorption variation across this range being negligible.

The alignment of the SPDC source was performed by fibre-coupling the frequency-filtered emission into single-mode fibres that were connected to SPADs. The correlated signal and idler modes were found by translating the fibre-coupler and optimizing the coincidences between detected photons using an electronic circuit. In the single-photon SPDC experiments, the stimulus block was generated by a single trigger pulse to the ultraviolet laser and 'successful' events were post-selected based on the EMCCD camera registering one and only one photon. Because of the non-unity mean photon pair per pulse rate (0.048 – corresponding to a ultraviolet laser power of $\sim 150$ μW) and additional EMCCD noise (clock-induced charge rate of $\sim 0.04$), an experimental session typically consisting of one thousand trials ($=$ trigger pulses) yielded $\sim 80$ single-photon events passing the post-selection. The optimal EMCCD operating settings were determined by a combination of the EMCCD camera's theory of operation, numerical simulations and experimental testing. The settings of the EMCCD camera as utilized in the experiment are given below as well as in Supplementary Table 2.

**Characterization and performance of the EMCCD camera.** In this work, the expected photon number per EMCCD pixel in the single-photon detection regime is $< 0.1$. Therefore, the application of a fixed count threshold, above which one photon is said to have been detected, typically yields best results[29]. We empirically determined the distribution of the EMCCD counts to be normal with an average of 200 and a s.d. of 3.5 counts by operating the camera in the absence of stimulus light and with its external shutter closed. Therefore, for our particular camera, we set the threshold as 220 EMCCD counts, which is the $\sim 6\sigma$ value above the readout noise (200 counts). Furthermore, the readout region of interest was a three-by-three pixel area and events were classified as single-photon events if this threshold was exceeded on one and one pixel only.

To minimize intrinsic noise events, the EMCCD camera (Andor iXon 897 Ultra) must operate in a continuous long series kinetic mode – this behaviour was not specified by the camera manufacturer but was noticed experimentally to produce the best results. On reception of a trigger pulse, the experimental control software waited for a signal that the camera had begun its next acquisition. As the output of the laser has a delay ranging between $\sim 35$ and $50$ μs following the reception of this trigger pulse, the exposure time of the camera was set to 100 μs to ensure synchronization between the camera and the laser. However, exposure time is a non-critical variable and can be increased without noticeable degradation of performance, as the dominant noise mechanism in this regime is clock-induced-charge, a mechanism entirely independent of the length of the exposure.

The most effective settings for the EMCCD camera were determined by a combination of the EMCCD camera's theory of operation, data collection and numerical simulations. The settings of the EMCCD camera when run in single-photon counting mode as utilized in the experiment are given in Supplementary Table 2.

The heralding efficiency, that is, the probability for a single photon impinging on the cornea provided its partner is detected by the EMCCD, of the SPDC source in combination with the EMCCD detection was estimated at $\sim 20\%$. This was determined based on the heralding efficiency measured directly with SPADs (coincidence to trigger single photon rate), but taking into account their detection efficiency ($\sim 40\%$) as well as the noise (clock-induced charge rate $\sim 0.04$) of the EMCCD. This value of the heralding efficiency was also consistent with numerical simulations.

**Elimination of two and higher number photon states.** One of the key advantages of using the single photon SPDC source in combination with a multi-pixel detection array such as an EMCCD in our study was that it allowed for the identification of cases in which two or higher number photon states were generated, which could then be excluded from further analysis through post-selection. This was possible because of the high quantum efficiency of the EMCCD

that we employed for detecting the idler photon and also because, unlike when conventional SPADs are used, the spreading of the spatial mode of the idler beam onto a three-by-three pixel region of the EMCCD allowed for the detection of trials in which two or multi-photon pairs were generated. At the SPDC pump power used in the experiment (laser power $= 150$ μW) which yielded a mean photon pair rate per pulse of 0.048, the rate of two and multi-pair events was already highly reduced ($\sim 0.11\%$). Moreover, using EMCCD in the above configuration allowed us to detect $\sim 80\%$ of such cases where two or a higher number of photons arrived at the EMCCD and such events were discarded during the post-selection. This overall ability of the EMCCD to identify two and multi-photon events, together with the efficiency of the signal arm, the transmission efficiency of the ocular medium and the quantum efficiency of the photo isomerization left us with only $\sim 0.02\%$ of post-selected trials in which two or more photon states were generated by the crystal, misidentified by the EMCCD camera as a single-photon event and elicited a two- or multiple-photon signal on the retina. This fraction of trials that lead to multiple-photon events was obtained from a stringent analysis of the $g^{(2)}(0)$ function (see above) and taking into account measured values for photon loss and detector inefficiencies in the signal and idler arms. This means that from our 2,420 trials out of $\sim 30,000$ that passed the post selection, on average $< 1$ multiple-photon event would have occurred at the level of the retina, which we feel can be safely ignored. Thus, unlike in previous studies we can exclude the possibility that the subjects' responses were due to two- or multiple-photon events.

**Heralding efficiency of the SPDC source.** A commonly used metric to quantify the quality of the single-photon source is the heralding efficiency. For a source based on SPDC this denotes the probability that a photon was present in the signal arm conditioned on the detection of the idler photon. For our set-up the heralding efficiency was $\sim 20\%$ (see above).

This imperfect heralding efficiency affected neither the ability of subjects to detect single photons nor the validity of our observation on single-photon-induced modulation of the sensitivity of the visual system. This is because even a low heralding efficiency would only lead to a higher number of blank trials but would not lead to more than a single photon being present at the subjects' cornea. Furthermore, our main results are in good agreement with the heralding efficiency ($\sim 20\%$) and the overall quantum efficiency of the eye estimated previously ($\sim 30\%$, see Supplementary Table 1). Based on these values, the maximum theoretical probability of correct response was estimated to be $\sim 0.53$ (all post-selected trials, see Fig. 2a, Supplementary Fig. 2). We also note that this calculation does not hold in case of dividing the post-selected trials into different ratings, for example, in case of the high-confidence rating R3 events, this data effectively already presents a further post-selection or subset of all experimental trials, in which with the high probability visual system indeed received and detected a single-photon event.

**Numerical simulations for required number of trials.** The SPDC single-photon light source and the optical set-up were simulated using the Monte Carlo method, including a simulation of the output produced by the EMCCD. In the simulation, SPDC photon pairs were pseudorandomly generated from a Poisson distribution determined by the mean number of pulse pairs. Experimentally measured transmission values for both arms (signal and idler) were used to obtain probabilities of the photon numbers incident on the eye and the EMCCD.

The idler photons of a pair that survived transmission through the set-up were pseudorandomly distributed onto EMCCD pixels based on the measured beam profile. A simulated number of EMCCD counts was generated for each pixel in the three-by-three region of interest, accounting for all the stages between detection of the photon and the electronic readout, including the stochastic gain mechanism. A Monte Carlo simulation was chosen to fully account for the effects of possible serial register clock-induced-charge. First, clock-induced-charges were accounted for by the pseudorandom addition of an electron to the photoelectron counts in the pixel. The combined photoelectrons and clock-induced-charge electrons are then carried through the EMCCD register with a probabilistic gain applied to each electron individually and at each register stage. The gain (G) is given by,

$$G = (1+p)^r \tag{1}$$

where $p$ is the probability of an electron producing a secondary electron as it is shifted through the register and $r$ is the number of stages in the register, which in our case was 512.

Additionally, a probabilistic serial register clock-induced-charge is also accounted for at every step. This probability was determined by matching the simulated data to the experimental count distribution. Finally, the total number of electrons is then converted into a readout number of counts, based on the experimentally observed distribution, to give an EMCCD count number for each pixel.

For each trial, the number of pixels with an above-threshold number of counts was then compared with the number of photons remaining in the signal arm. Repeating this multiple times allowed for the generation of the expected photon number distribution impinging on the eye for different detected photon numbers at the EMCCD.

The number of experimental trials required was estimated by using the probability of correct response ($\sim 0.516$) based on an ideal detector limited by physiological estimates of noise and efficiency and the simulated fraction of events

passing the post-selection for the different set-ups. The necessary number ($n$) of single-photon trials (that is, post-selected trials) required to bring the expected probability of correct response to above chance at the 95% confidence level ($z = 1.96$) was then calculated by,

$$n = z^2 \frac{p \cdot (1 - p)}{(p - \mu)^2} \qquad (2)$$

where $p$ is the expected probability of correct response and $\mu$ is the value at random chance (that is, $\mu = 0.5$). The total number of trials that would need to be performed to reach this many post-selected events as a function of source and power could be calculated by simply dividing by the expected fraction of events passing the post-selection criteria.

Similarly, the number of required experimental trials when using an SPAD as an idler (trigger) detector was calculated, with the SPAD's binary detection modelled by a binomial trial based on their known efficiency.

Furthermore, the expected $g^{(2)}(0)$ values were also simulated analogously (solid lines in Supplementary Fig. 1). We simulated propagation and simultaneous detection of signal photons in both arms behind a 50/50 beamsplitter, together with coincident detection of a photon in the idler arm.

**$g^{(2)}(0)$ correlation function measurement.** Attenuated continuous wave (CW) laser light and the Fock states of SPDC have different photon number distributions, which can be measured and distinguished with the second-order quantum correlation function $g^{(2)}(0)$ (ref. 30), which relates the mean photon number to its s.d.:

$$g^{(2)}(0) = 1 + \frac{< (\Delta n)^2 > - < \hat{n} >}{< \hat{n} >^2} \qquad (3)$$

where $\Delta n$ is the s.d. on the photon number $n$, and $< \ldots >$ represents the average. It is evident from equation (3) that in the case of Poisson distributed coherent states, in which the variance is equal to the mean (that is, $< (\Delta n)^2 > = < \hat{n} >$), $g^{(2)}(0) = 1$ and that for a one photon Fock state with zero variance, $g^{(2)}(0) = 0$. The second-order quantum correlation function thus represents a good and universally accepted test for the quantum nature of a light source, especially as it can be measured using a simple Hanbury Brown and Twiss interferometer[31]. To do this, we fibre-coupled the SPDC signal emission into a single-mode fibre-beamsplitter (Thorlabs) whose outputs were directly connected to two additional SPADs. The idler arm fibre was also detected with an SPAD. In this configuration, the $g^{(2)}(0)$ measurement reduces to:

$$g^{(2)}(0) = \frac{N_{c3} N_i}{N_{s1} N_{s2}} \qquad (4)$$

where $N_i$ is the number of counts in the idler arm, $N_{s1}$ and $N_{s2}$ are the counts at the two detectors in the signal arm coincident with an idler count, and $N_{c3}$ is the number of triple-coincidences between the idler arm and both arms of the signal. Our coincidence window was set to the laser pulse duration (10 ns), as all intra-pulse pairs are indistinguishable in the final experimental configuration.

We measured the $g^{(2)}(0)$ value of our SPDC source with both an SPAD and the EMCCD camera in the idler (trigger) arm. The results of both measurements are depicted in Supplementary Fig. 1, and show the expected dependence on the mean photon pair per pump pulse, which in turn is a function of pump laser intensity. The experimental results are in line with numerical simulations taking into account the parameters and theory of operation of both the SPAD and EMCCD detectors[29]. These numerical simulations also allow for the comparison of the ratio of multiple-to-single-photon events passing the post-selection criteria for the EMCCD and SPAD configuration respectively (Fig. 1b).

Most importantly though, using an EMCCD results in a high increase in triggered and usable single-photon events, due to the improved quantum efficiency (95% versus $\sim 40\%$) of the EMCCD compared with the SPAD, the lack of additional fibre-coupling losses ($\sim 60\%$) and the EMCCDs partial ability to identify and reject cases in which more than one photon had been emitted in the idler arm (multi-pair emission), due to its pixel-array structure.

**Poisson light source set-up.** We used a classical Poisson light set-up for our control experiments at high photon numbers and for training purposes. A schematic showing a simplified set-up of the CW experiments is shown in Supplementary Fig. 4a. The Poisson light source used in this work was a multi-line continuous wave argon ion laser (LASOS). A filter (Semrock TBP01-501/15) in combination with an acousto-optical modulator (AOM; AA Opto-Electronic) were used to select wavelengths around the peak wavelength of human rod cell's absorption curves between 495 and 505 nm, with the dominant line of the laser (488 nm) being filtered-out by this method. Stimulus pulses were sent to the subject's eye by activating the AOM to deflect light into a single-mode fibre. The deflection period was 1 ms and the voltage applied to the AOM modulated the power of the deflected and hence coupled light. The output of this single-mode fibre was collimated and focused on the pupil of an eye using a lens with a focal length of 750 mm. Neutral density filters further decreased the power of the light to the desired low-photon regime. A 50/50 beamsplitter was placed in between the focusing lens and the subject's eye, to allow the experimental photon levels to be monitored by coupling the non-experimental arm of the split light into a multi-mode fibre, whose output was attached to an SPAD (Perkin-Elmer

SPCM-AQR-14-FC). The output of the SPAD was gated, and only counts simultaneous with the light being presented (1 ms). However, as the dark count rate of these particular SPADs is $\sim 10^2\,\mathrm{s}^{-1}$, there was a mean dark count signal of $\sim 0.1$ per trigger period. No background-subtraction was performed.

The experiments were performed at seven different mean photon numbers from 20 photons with increasing steps of 20 up to a mean of 140 photons. Individually characterized neutral density filters and a low intensity power metre were used to set these photon number levels. The AOM was calibrated before each experimental session to set the precise number of photons at the cornea. Photon statistics were collected using the aforementioned SPAD for different voltages applied to the AOM to obtain a calibration curve. We ensured that for each voltage the distribution followed Poissonian statistics. Using the obtained calibration curve and known efficiency of the SPAD, the AOM voltage was set accordingly to achieve the desired photon numbers during the experiment. This was also confirmed by simultaneously recording the photon numbers in the non-experimental arm of the 50/50 beamsplitter with the SPAD during all experimental trials. Unwanted additional background light was tested for by observing both the background count rate of the SPAD in the experimental set-up and when the detection region of the SPAD was completed blocked from light. No significant difference in background level was observed.

Several sessions a week were conducted for each subject. Each session consisted of 165 semi-randomly scrambled blocks each containing five trials of the same light intensity. The ordering of the presentation of each light intensity was pseudorandomly shuffled by the computer control programme at the beginning of each experimental session without the subject or the experimenter having any knowledge of this order. Each light level was then presented in five consecutive trials, before the next light level was presented. The subject was informed that the level of the presented light was to change by a slightly prolonged beep through headphones. The different light levels were looped through in this manner until the required number of trials at each light level for that session had been performed, generally 75 per photon number per session.

**Statistics.** Unless stated otherwise, $P$ values are stated as the Fisher exact test.

**Mathematical modelling.** Detailed description of the mathematical model based on the Signal Detection Theory that we used to fit our data is presented in the Supplementary Note 3. Parameters of the model obtained by fitting the data are shown in Supplementary Table 3.

**Data availability.** The data supporting the findings of this study is available from the corresponding author on request.

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

## Acknowledgements

We thank M. Colombini and the IMP mechanical workshop for design and manufacturing of equipment used in this study, R. Ursin and A. Zeilinger for loan of counting electronics, M. Arndt for loan of a spectrometer and A. Straw and G. Tkačik for comments and suggestions. We would also like to thank A. Stark, J. Brennecke, B. Roska, M. Shadlen and J. Hudspeth for their careful reading and critical comments on an earlier version of the manuscript. M.M. and R.P. acknowledge the VIPS Program of the Austrian Federal Ministry of Science and Research and the City of Vienna. R.P. was supported by the European Commission (Marie Curie, FP7-PEOPLE-2011-IIF). The research leading to these results has received funding from the Vienna Science and Technology Fund (WWTF) project VRG10-11 and LS14-009, the Human Frontiers Science Program Project RGP0041/2012, the Research Platform Quantum Phenomena and Nanoscale Biological Systems (QuNaBioS) and Research Institute of Molecular Pathology (IMP). The IMP is funded by Boehringer Ingelheim.

## Author contributions

J.N.T., M.I.M., R.P. and A.V. designed and performed the experiments; D.W., J.E.-P. and M.L. participated in the study; J.N.T., M.I.M., R.P. and A.V. wrote the manuscript, A.V. conceived and supervised the study.

## Additional information

**Competing financial interests:** The authors declare no competing financial interests.

