## [Peer Review File · Nature Communications]

Reviewers' comments:

Reviewer #1 (Remarks to the Author):

Iconic, classical experiments ought to be repeated every now and then, both to replicate the original results, as well as to see whether with the intervening scientific and technological progress, additional information can be learnt. The group of Alipasha Varizi has attempted to do this with the 1942 experiment of Hecht, Schlaer and Pirenne from Columbia University (in a beautiful 21 page manuscript; indeed, overall it's a sad testament to the state of the publishing industry that the 1942 manuscript is a pleasure to read, more than 70 years after its original publication, as it is not marred by constant need to jump around to catch some supplementary experiment or figure. All control experiments and considerations are contained within a single narrative flow, unlike this and other contemporary manuscripts). Hecht et al showed that dark-adapted human subjects can detect flashes of light that contain as few as 5-8 quanta of 510 nm (blue-green) light at the rod photoreceptor level. As Hecht and colleagues used a classical light-source, photons could not be accurately counted as they were limited by Poisson statistics.

In collaboration with the Zeilinger and Arndt groups in Vienna - pioneers in advanced quantum metrology tools and instrumentations - Tinsley et al tried to redress this weakness of the Hecht et al experiment by using quantum optical light sources that enable the reliable generation (and subsequent registration) of single photons (by detecting the second in a pair of entangled photons) at the desirable wavelength within a temporal two-alternative forced-choice paradigm, in which subjects also had to signal the confidence in their judgment using a ternary scale. The physical apparatus, the control experiments & simulations to ascertain the generation of single photons (Fig. 1) and the psychophysical setup are all superbly designed and executed. A very difficult and trying experiment (because <10% of the trials included photons; that is, the 3 subjects effectively made random decisions for ca 28,000 trials and had to assign a confidence rating to these trials. oh vey!) that was a pleasure to read.

I applaud the authors for carrying out these experiments. However, as detailed below, the data themselves are not nearly as conclusive as the authors claim. The strong claim made in this manuscript - humans can detect the absorption of a single photon in a single rod - requires strong proof. This is not it. Thus, the authors should cut such phrases as "This demonstrates that subjects can indeed detect a single photon with a probability above chance."

However, even this limited claim is quite extraordinary; to wit that a classical system such as the retina, a piece of wetware at room-temperature, can detect a single quantum of visible light! The experiment deserves a wide readership.

I would be more than happy to re-read the next iteration of this paper

Specific comments -

The heart of the data is contained in Fig 2. Of the 2,420 single-photon events, the average detection probability was 51.6+/- 1.0 %. This result is quite weak, with an associated p value of 0.0545. This low value is not further discussed. This won't do. In behavioral experiment of this sort, with myriad of variables (including very difficult factors to control for such as different genetic background, different sleep-wave cycles, arousal, more or less tea, coffee or other stimulants during the experiments and so on) requires a more stringent significance threshold (I personally prefer a 0.01 significance level). This is a statistical trend but no more.

How does the statistic look for the three individual subjects? Do any of them exceed significance?

However, a somewhat more restricted claim of "subjects that are very confident in their response (R3) can detect single photons better than chance" appears to be valid (Fig. 2a) with a detection performance of 60%+/-3%, which is highly significant above chance ($p=0.001$).

However, in only 242 out of 2420 trials (exactly 10%) did subjects report high confidence. So it's a small number out of the 30,000 overall trials. Could the authors break down these 242 trials for the three individual subjects? Do they have sufficient statistics to claim that every subject performed above chance when they were sure of their response? Do they occur bunched in time (i.e. early in the experiments but not late when subjects were tired etc?)

Are the data presented in Fig 2b for all responses or only for correct responses (this is what should be shown).

The authors subsequently identify a priming effect; that is, after one photon has been sent into the eye, the next photon is more likely to be detected within the 5 or so seconds (Fig. 2c,d). Again, given the low performance, 51.0% +/-1.1% (no significance is provided), I very much doubt it. As above, when subjects were confident of their responses, the effect seems real (0.59+/-0.04 with $p=0.02$). Please modify the text accordingly.

For Fig. 2d, in which the authors estimated the excess detection probability as a function of the time from the preceding single photon event, it is questionable whether any of these data are significant different from chance (here 0%). Each of these points ought to have a significance value attached to it.

I understand that these are rather demanding experiments. However, the claims that this manuscript makes are very strong and must be adequately supported. I do believe their data demonstrates that on the small fraction of trials in which people are very confident of their responses (R3), subjects detected a single photon event.

This raises the question of post-retinal factors that are responsible for the difference between R1, R2 and R3 trials. What makes the difference between low and high confidence detection. This could relate to a slew of factors that psychophysicists have investigated for decades including habituation, fatigue, arousal, selective visual attention, suppression of tremor or microsaccades, or even subtler phenomena

such as relate to the heart beat, phases of the brain's EEG in the beta range etc. Food for future experiments. However, such non-retinal factors ought to be at least discussed.

Reviewer #2 (Remarks to the Author):

The authors present a novel light source that give them some control and knowledge over the number of photons delivered to the eye in a single flash. They use this light source to generate single photon stimulations to the eye. They find that a human's ability to detect a single photon stimulus is just a hair above chance, albeit significantly so.

The originality of this paper is in the light source, and the fact that it is used to address long-standing fundamental questions about human vision.

I do not necessarily doubt that it is possible for a human to sense a single photon. Extrapolation of psychometric functions from Hecht, Schlaer and Pirenne's 1942 paper suggests this possibility. However, the finding that the ability to detect a single photon is so slight, that even the slightest error or oversight in the analysis might lead to an erroneous result. Given that the authors are making such a profound statement, they owe it to the scientific community to be more convincing of this fact.

Finally, many question that arise from read in the main text are only answered by reading the Supplemental Materials. This may not be easy to fix, except by publishing the paper in a journal that allows for all important information to be in the main text.

Specific Comments:

Page 4: Is it possible that the EMCCD detected a single photon when in fact there were two? Were the authors 100% certain when they selected the 'single photon trials'? It is not clear, even from the supplementary materials, that this is possible. According to Fig 1b, the detectors themselves record different numbers of photons, so one or both of the detectors are not 100% accurate. It is possible, therefore that some of the 'one photon' trials had more than one photon sent to the eye.

Page 4 para 2: replace 'less' with 'fewer'

Page 5, para 2: The authors report 2420 single photon events out of 30767 trials. This is 7.8 percent of trials, which is way more than the intended percentage of 4.8 reported in the previous page. Can the authors explain this?

Page 5: The chance of reporting the interval containing the photon is just a hair above chance. The authors should be clearer in how they report this number. A reader might misinterpret the numbers

reported as stating that subjects have just over a 50% chance of seeing a photon that strikes the eye, which is far from the case.

Page 9, last para: It seems very unlikely that EEG or MEG will contribute anything to understanding this phenomenon. It's recommended to remove this section.

Page 11, line 1. "...barely visible red light...". The control of the light levels is critical for this experiment, so it is surprising that the authors provide no details, or express no concern over the light used for fixation. What is the wavelength of the fixation light? What is the power? What is the bandwidth? Given typical levels of light scattering in a human eye, what number of photons from this source will land on the tested area that is 23 degrees away? Given the spectral sensitivity of the rods, what is the probability that these photons might be seen?

Page 11: para 2, line 1: Who triggered each trial? Was it the subject (ie self-paced)? What was the average time taken for each trial.

Page 11: para 2, last line: Why was feedback used? This is a crucial issue here. It makes sense that feedback was used for training (Suppl. Mat), but not for the experiment. With feedback, it is possible that the subjects could have used other cues, however subtle, to know when a photon was emitted. Did the experimenters try the experiment with no feedback? If yes, then what were the results? If not, then they should.

Page 11, para 4: 1000 trials over two hours is 7 seconds per trial. If the 2 hours includes the ~ 1 hour of dark adaptation, then the average time per trial was 3.6 seconds. Given the long time between trials, how can the authors get a data point for probability of a correct response at 2 seconds between the current and preceding event (Fig 2(c&d))?

Page 12, bottom: Contrary to what is written in the methods, it appears from figure 1 that there is no fiber in the Idler arm but that there is a single mode fiber in the Signal arm. What are the coupling losses of this fiber? My understanding is that they are about 75% at best. It seems that there may be more events where a photon was detected by the EMCCD but its paired photon never makes it to the eye.

Page 13, para 2: Can the authors be certain that '..one and only photon..' events are determined accurately by the EMCCD? Is it possible that a multiple photon event would be mis-identified as a single photon event?

Figure 2: What are the error bars?

Supplementary Table 1: Please specify the wavelengths that these numbers refer to.

Reviewer #3 (Remarks to the Author):

While the sensitivity of human eyes to light has been determined to be on the few photon level a few decades ago in a number of celebrated experiments. The absolute lower limit was not known. The use of a pair of correlated photons, the authors develop a novel experimental scheme that they can ensure that they can identify trials where only a single photon was incident up the retina with unprecedented accuracy (approximately an order of magnitude better than traditional Poisson noise). This is a very novel application of a correlated photon source. The exciting result is that our eyes are responsive to single photon with probability slightly higher than random. It is also interesting that they find a "potentiation" effect where detection of one photon increases the probability of future detection within some time constant. Overall, the experiments are well designed with appropriate simulation and theoretical estimate to bolster the experimental results. The paper is well written and clearly presents these original findings. I have only a few minor comments:

1. While it is convincing that R3 responses have a probability of 0.59 ± 0.04 , the combined responses have only a probability of 0.51 ± 0.01 . Unless I am missing something, it does appear that the combined probability is above random chance.
2. Please explain why if read noise of EMCCD is 200 count and 220 count corresponds to 6 sigma. What is the underlying statistic of this 200 count read noise (it is not Poisson for sure. Does the 200 count include some pedestal value?)
3. The use of EMCCD as a large number of parallel detectors to ensure multiphoton events can be distinguished from single photon events is an important innovation in the experiment. It is expected that it should be improved upon single pixel detectors like SPAD but does this improvement depend on the correlated photon generation geometry and the intermediate optics between the photon source and the EMCCD?

Remark to all reviewers

We would like to thank all the reviewers for their critical reading of our manuscript. We were delighted to see the extremely positive and enthusiastic assessment of the reviewers, e.g. “The experiment deserves a wide readership”, “ ... the novel light source is used to address long-standing fundamental questions about human vision”, “This is a very novel application of a correlated photon source”, “the experiments are well designed with appropriate simulation and theoretical estimate to bolster the experimental results. The paper is well written and clearly presents these original findings”, “The physical apparatus, the control experiments & simulations to ascertain the generation of single photons and the psychophysical setup are all superbly designed and executed”, “a very difficult and trying experiment [...] that was a pleasure to read. I applaud the authors for carrying out these experiments”.

We also found the referees’ constructive suggestions very valuable and helpful in improving the overall quality of our manuscript. In our letter below we provide, among other things, new experimental data (Figures R1.1 and Figure R2.1) that further corroborates our initial finding of the detection of a single photon by humans. We have also performed new data analyses on subject performance (new Supplementary Figure 5, Figure R1.2, Figure R1.3 and Figure R3.1), and provided control data as well as a more comprehensive and detailed description of the used protocols, calibrations and properties of the single-photon source (revised Methods and revised Supplementary Note 4). We have additionally revised Figure 1, Figure 2 and Supplementary Table 1 as well as made essential corrections and extensions to the original main text of the manuscript (see unlined changes in the revised manuscript).

In addition, we are pleased to report that we have been able to address all the comments from all the reviewers in full, as detailed in our point-by-point response below.

Reviewers' comments:

Reviewer #1 (Remarks to the Author):

Iconic, classical experiments ought to be repeated every now and then, both to replicate the original results, as well as to see whether with the intervening scientific and technological progress, additional information can be learnt. The group of Alipasha Varizi has attempted to do this with the 1942 experiment of Hecht, Schlaer and Pirenne from Columbia University (in a beautiful 21 page manuscript; indeed, overall it's a sad testament to the state of the publishing industry that the 1942 manuscript is a pleasure to read, more than 70 years after its original publication, as it is not marred by constant need to jump around to catch some supplementary experiment or figure. All control experiments and considerations are contained within a single narrative flow, unlike this and other contemporary manuscripts). Hecht et al showed that dark-adapted human subjects can detect flashes of light that contain as few as 5-8 quanta of 510 nm (blue-green) light at the rod photoreceptor level.

As Hecht and colleagues used a classical light-source, photons could not be accurately counted as they were limited by Poisson statistics.

In collaboration with the Zeilinger and Arndt groups in Vienna - pioneers in advanced quantum metrology tools and instrumentations - Tinsley et al tried to redress this weakness of the Hecht et al experiment by using quantum optical light sources that enable the reliable generation (and subsequent registration) of single photons (by detecting the second in a pair of entangled photons) at the desirable wavelength within a temporal two-alternative forced-choice paradigm, in which subjects also had to signal the confidence in their judgment using a ternary scale. The physical apparatus, the control experiments & simulations to ascertain the generation of single photons (Fig. 1) and the psychophysical setup are all superbly designed and executed. A very difficult and trying experiment (because <10% of the trials included photons; that is, the 3 subjects effectively made random decisions for ca 28,000 trials and had to assign a confidence rating to these trials. oh vey!) that was a pleasure to read. I applaud the authors for carrying out these experiments. However, as detailed below, the data themselves are not nearly as conclusive as the authors claim. The strong claim made in this manuscript - humans can detect the absorption of a single photon in a single rod - requires strong proof. This is not it. Thus, the authors should cut such phrases as "This demonstrates that subjects can indeed detect a single photon with a probability above chance." However, even this limited claim is quite extraordinary; to wit that a classical system such as the retina, a piece of wetware at room-temperature, can detect a single quantum of visible light! The experiment deserves a wide readership.

We thank the reviewer for the positive assessment of our work and we are glad to see that they consider our results worthy of a wide readership. We agree with the reviewer that some of the claims in our initial manuscript may not have been fully supported by the presented data. As outlined below in detail we have addressed this issue by toning down such claims in the current version of the manuscript, as well as providing new data – to the extent that was possible to collect these within the available time frame for the revision – which provide corroborative support for our initial experiments. We hope these additions and clarifications will fully address all the reviewer’s initial concerns.

I would be more than happy to re-read the next iteration of this paper

Specific comments -

The heart of the data is contained in Fig 2. Of the 2,420 single-photon events, the average detection probability was 51.6+/- 1.0 %. This result is quite weak, with an associated p value of 0.0545. This low value is not further discussed. This won't do. In behavioral experiment of this sort, with myriad of variables (including very difficult factors to control for such as different genetic background, different sleep-wave cycles, arousal, more or less tea, coffee or other stimulants during the experiments and so on) requires a more stringent significance threshold (I personally prefer a 0.01 significance level). This is a statistical trend but no more.

We agree with the reviewer that a p-value of $p=0.0545$ is not a very stringent significance threshold. We also agree that given the large number of variables highlighted by the reviewer that, if considered on its own and irrespective of the other results presented in the manuscript, the probability of correct response for the combined confidence ratings may not be fully convincing. However, as the reviewer kindly acknowledges these experiments are extremely demanding. Obtaining a p-value of 0.01 over all responses as

suggested by the reviewer requires the performance of approximately 45,000 additional trials, far more than all the data collected over the course of several months taken for this study. Therefore, obtaining such an additional, large dataset merely to improve the p-value would not be feasible within the available time for a revision and may also not, as we believe, be strictly necessary to support the conclusions of the current work.

We have addressed this concern by providing the following additional experimental data and lines of support:

1. Instead of the more time-consuming single-photon SPDC data, which as mentioned would not have been feasible within the timeframe for a review, we deferred to experiments in which subjects were presented with pulses containing a Poisson-distributed average photon number of one. In such experiments significantly fewer blank pulses are generated on average and it was thus possible to perform a sufficient number of experimental trials within a few weeks. We performed an additional total of 2,780 trials with 4 subjects (the 3 original Subjects A-C and an additional Subject D), which yielded a combined probability of correct response of 0.517 +/- 0.01 (p-value 0.036) (see Fig. R1.1 below). Of course, experiments with attenuated Poisson-distributed light cannot directly prove the detection of single photons by human subjects due to their inherent variance in photon number. This limitation was in fact the original motivation for the development of our SPDC source. Nevertheless, the fact that the data from these experiments is consistent with the results obtained with our single-photon quantum light source, gives corroborative evidence for our single-photon detection data. We feel this is the best we could do in terms of additional experiments within the available timeframe.

Figure R1.1. Probability of correct response in quantum light source single-photon experiments (original data, 3 subjects) and Poisson light source with an average photon number one (new data for 4 subjects). The probability of correct response is shown for combined ratings (brown bar) and the high-confidence R3 ratings (green bar) together with corresponding p values as well as the Fisher combined p values.

Importantly, for a Poisson source at an average photon number of one and considering the ~30% efficiency of the visual system (Supplementary Table 1), the probability that two or more photons lead to light induced, multiple photon events at the retina is ~ 3.7%. Given the expected small contribution of such multiple-photon events compared to the cases where a single photon was sent to the eye, we reasoned that both sets of experimental data could be used to test the same hypothesis. We could therefore improve the overall p-value by combining the evidence with the original dataset using Fisher's method [Fisher. 1932. Statistical Methods for Research Workers]. The data collected from the SPDC and the classical light combined in this way yield a much more significant p-value of 0.014, compared to our original value of 0.0545.

2. Aside from the above new data, we also believe that there are at least two other facts that further support our conclusion that subjects can distinguish single photon flashes:
 - a. First, as the reviewer correctly stated in the next point, when only high confidence R3 ratings are considered our p-value is indeed significant ($p=0.001$). Thus, based on this observation we believe it is possible to claim that in the small fraction of trials in which people are very confident of their responses (R3), subjects detected a single photon.
 - b. Second, it is the correlation between the indicated high confidence ratings *and* the higher probability of correct response for these trials that provides additional support that subjects can indeed see a single photon. It's important to realize that such a correlation *per se* does not necessarily have to exist.
Based on the number of true single-photon events, i.e. when a single photon was delivered to the retina, and the reported values for the efficiency of the eye (~30%, Supplementary Table 1) we expect that only in ~ 6% of the post-selected trials an actual light induced signal was generated in the retina. Therefore, it is expected that only this fraction of trials should be able to contribute to an above chance performance as well as to some increase in the subjects' choice of high confidence (R3) ratings. This congruence suggests that in the majority of the trials, in which the impinging photon onto the eye made it to the retina and generated a light-induced signal the subjects are more likely to chose the high confidence R3 rating. This notion is further supported by the fact that in the combined low confidence trials (R1 and R2) the subjects' performance was never above random chance as evidenced by a p value of 0.23 (R1 and R2 combined, $n = 2178$).
3. Finally, we developed a comprehensive mathematical model based on Signal Detection Theory (SDT) that put all our experimental observations, including the probability of single photon detection, the distribution of ratings and the single-photon induced priming effect, within a single framework. It assumes that every single photon incident onto the eye has a certain probability of being detected by the subject. The model takes into account known physiological values of the

efficiency of the eye (Supplementary Table 1), the temporal characteristics of the gain extracted from the experiments (Fig. 2c) and the two decision criteria used by the subjects to assign their ratings (Fig. 3a-b). We show that our model correctly predicts both the observed distributions of ratings and the probability of correct response at each rating for the single photon regime (Fig. 3d-f). We believe that taken together these further supports the claim that subjects are indeed capable of detecting single photons.

We have reworded the manuscript (see underlined changes on pages 5 - 7) in order to reflect these aspects. We also acknowledge the lower statistical significance of the $p=0.0545$ value, as pointed out by the reviewer and have correspondingly adjusted our statements (underlined changes on page 5). Overall, we hope that we have been able to sufficiently satisfy the reviewer's concerns with these measures.

How does the statistic look for the three individual subjects? Do any of them exceed significance?

Following the reviewer's suggestion we have looked at the statistic of the data for the individual subjects and for combined ratings. As such a breakdown of data leads to a corresponding reduction of the number trials for the statistical analysis, we found that under these conditions the corresponding p-values were not statistically significant for the individual subjects' combined ratings. Although this suggests that no significant conclusions about the ability of specific individuals to discriminate single photons can be made, pooling the data allowed us to obtain sufficiently high number of trials that was essential to obtain the main conclusions of the work with statistical significance. We have therefore chosen to not include and draw any conclusions from the individual subjects' data in our revised manuscript.

However, a somewhat more restricted claim of "subjects that are very confident in their response (R3) can detect single photons better than chance" appears to be valid (Fig. 2a) with a detection performance of $60\% \pm 3\%$, which is highly significant above chance ($p=0.001$).

We fully agree with the reviewer. As discussed earlier and highlighted in the revised manuscript we have toned down our wording about the conclusions that can be drawn from the combined rating analysis and have adjusted our statements correspondingly about the high confidence R3 rating (underlined changes on page 5-6).

However, in only 242 out of 2420 trials (exactly 10%) did subjects report high confidence.

So it's a small number out of the 30,000 overall trials. Could the authors break down these 242 trials for the three individual subjects? Do they have sufficient statistics to claim that every subject performed above chance when they were sure of their response?

We thank the reviewer for the suggestions. Following the reviewer's comment, we broke down all high confidence trials into individual subjects. We found that although for all subjects the probability of correct response showed a similar trend (Subject A: 0.64 ± 0.1 , Subject B: 0.6 ± 0.05 , and Subject C: 0.57 ± 0.09), given that such a breakdown of

our original dataset reduces the number of trials for the statistical analysis, we expectedly found that only one subject (B) performed at a statistically significant level above chance. Therefore, the pooling of data across subjects was essential in obtaining a sufficient number of trials necessary to support our conclusions with statistical significance. Thus, as already mentioned above, we have chosen not to include or draw any conclusions from the individual subjects' data sets in our revised manuscript.

Do they occur bunched in time (i.e. early in the experiments but not late when subjects were tired etc?)

Following the reviewer's suggestion we have also analyzed the timing distribution of correct high confidence responses. Our new Supplementary Fig. 5a shows that the time between correct high confidence R3 responses is well described by an exponential distribution (R square = 0.98) demonstrating that these correct events occur randomly in time, as expected.

In addition we have plotted the distribution of times when correct high confidence R3 responses occurred during sessions (Supplementary Fig. 5b). The resulting distribution is not statistically significant from a uniform distribution ($p=0.07$, Kolmogorov-Smirnov test), demonstrating that the performance of the subjects is approximately constant and that subjects performed equally well throughout the whole duration of the session.

Are the data presented in Fig 2b for all responses or only for correct responses (this is what should be shown).

The original figure panel showed the distribution for all responses. We followed the reviewer's suggestion and have modified the graph accordingly which now shows the distribution of confidence ratings for correct responses only.

The authors subsequently identify a priming effect; that is, after one photon has been sent into the eye, the next photon is more likely to be detected within the 5 or so seconds (Fig. 2c,d). Again, given the low performance, 51.0% \pm 1.1% (no significance is provided), I very much doubt it. As above, when subjects were confident of their responses, the effect seems real (0.59 \pm 0.04 with $p=0.02$). Please modify the text accordingly.

We are fully aligned with the reviewer. The reviewer is probably referring to Fig. 2c, which shows that single photon detection is enhanced by a previous photon. Based on the 51.0% performance figure cited by the reviewer we assume she/he is referring to the region of the plot where events with long delays (over 10 s) are shown. We agree that under these conditions the performance is not significantly above chance (p -value = 0.2). However, our hypothesis of the single-photon induced priming effect is supported by the fact that for all combined-rating data at shorter delays (< 10 s) the observed probability of correct response is both elevated (0.56 ± 0.03) and significantly above chance (p -value = 0.02). Furthermore, and in agreement with the reviewers comment, the priming effect is statistically significant (p -value = 0.001) for high confidence data (Fig. 2c inset) and remains so even for longer delays (p -value = 0.02). This can be clearly seen when we pool across long delay high-confidence data (Fig 2c inset, right-hand plot). We have

followed the reviewer's suggestion and modified the text accordingly and state these observations more clearly in the revised manuscript (underlined changes on page 7).

For Fig. 2d, in which the authors estimated the excess detection probability as a function of the time from the preceding single photon event, it is questionable whether any of these data are significantly different from chance (here 0%). Each of these points ought to have a significance value attached to it.

The reviewer has correctly pointed out that most of the individual points in Fig. 2d are not significantly different from 0%. However, the point of this panel was to show that our model based on Signal Detection Theory (Supplementary Note 7), which takes into account the photon-induced priming effect, shows a similar trend at small delays as our data and provides an overall better fit to the data (shown as a solid line) than the null hypothesis, as represented by a horizontal, flat line at 0%. The χ^2 value for the fit of our model to the data (solid line in Fig. 2d) is 2.7 versus 9.6 for a flat line at 0%.

Moreover, while it is true that most individual points in Fig. 2d are not significantly above 0%, the p-value for the points combined on the interval 3 – 6 seconds is 0.007 and therefore the data in this range is significantly above 0%. This provides additional support for the suggested photon induce priming effect even when all confidence ratings are used.

We have made corresponding modifications to the main text to make these points more clear (underlined changes on page 22).

I understand that these are rather demanding experiments. However, the claims that this manuscript makes are very strong and must be adequately supported. I do believe their data demonstrates that on the small fraction of trials in which people are very confident of their responses (R3), subjects detected a single photon event.

We thank the reviewer for the encouraging and useful comments. We agree that one of the main conclusions of our work is that subjects can detect single photons in cases where they are confident in their response. As discussed above the frequency of these cases is consistent with what would be expected based on the detection efficiency for the signal photon in our setup and how often a single photon is expected to make it through the ocular medium and induce a signal at the retina.

Moreover, all our experimental data is fully explained by and is consistent with our model based on signal detection theory (Fig. 3d-f, Supplementary Note 7). This provides an additional support to our main conclusions.

We agree that some of our initial statements may not have been fully supported by the data presented. We have modified the text to reflect the reviewer's comments in this respect and made the above points more clear.

This raises the question of post-retinal factors that are responsible for the difference between R1, R2 and R3 trials. What makes the difference between low and high confidence detection.

The reviewer has raised again an important point for which we would like to thank her/him. A standard framework that allows the description of psychophysics data in which subjects make decisions and assign confidence ratings is provided by Signal Detection Theory (SDT) [Green and Swets, 1989, Signal Detection Theory and Psychophysics]. We provide a comprehensive mathematical model based on SDT that quantitatively explains our data (Fig. 3a,b, Supplementary Note 7) and puts all our experimental observations, including the probability of single photon detection (Fig. 2a), the distribution of confidence ratings and the single-photon induced priming effect (Fig 2b-d), within a single framework.

In our case SDT assumes that subjects obtain retinal signals during the two intervals of the 2AFC trial. It further assumes that the difference signal from the both time intervals is calculated through further processing and is compared against an internal criterion in order to assign a rating value. Large difference signals (above a certain criteria) are classified as high confidence, as it implies that during one of the intervals, the signal was significantly stronger than in the other, suggesting the presence of light. No or small differences is classified as low confidence (Supplementary Note 5 and 7). Our model also takes into account known physiological values of the efficiency of the eye (Supplementary Table 1) and the temporal characteristics of the gain extracted from the experiments (Fig. 2c).

We show that our model correctly predicts both the observed distribution of ratings and the probability of correct response at each rating for the single photon regime (Fig. 3d-f), and, therefore, accounts for and rationalizes the difference between low (R1, R2) and high confidence (R3) trials.

This could relate to a slew of factors that psychophysicists have investigated for decades including habituation, fatigue, arousal, selective visual attention, suppression of tremor or microsaccades, or even subtler phenomena such as relate to the heart beat, phases of the brain's EEG in the beta range etc. Food for future experiments. However, such non-retinal factors ought to be at least discussed.

We agree with the reviewer that the factors used by the subject in the assignment of the above criteria for the different confidence ratings could be influenced by a range of factors including those listed by the reviewer. Therefore, we have tried to make every possible effort to minimize or normalize for such factors and also achieve maximum sensitivity of our subjects. Mainly, we allowed subjects autonomy on all aspects regarding the pace of the experiments, in particular by placing the triggering of trials, the times taken to indicate their interval and confidence responses under their control. Additionally, subjects were told to take regular breaks during the experiments as needed and to stop if they felt tired, over-exerted or generally restless. Furthermore, it was the subjects who arranged when an experimental session would occur, a decision usually made at most a few hours in advance, and they were encouraged only to do so when they felt fit and able to do so to the best of their ability and concentration.

Furthermore we have performed additional analysis providing explicit control data for a range of factors including some of the specific points the reviewer has raised.

1. In order to exclude the influence of cognitive effects associated with the feedback subjects received on the previous trial we have analyzed separately events for which subjects received negative and positive feedback and found no correlation between the value of the feedback (i.e. correct or incorrect) and the probability of correct response. Nor did we find a correlation between the value of the feedback and the distribution of confidence ratings (Supplementary Fig. 6).
2. We further investigated if any cognitive effects relating to the subject's choice of their confidence rating may influence their performance in the *subsequent* trial. We found that whether subjects gave a high confidence R3 response or a low confidence R1 or R2 response does not significantly affect their probability of correct response in their subsequent trial. This suggests that any cognitive effects related to the subject's choice of their confidence rating, if at all present, do not significantly affect performance. Fig. R1.2 below illustrates this fact

Figure R1.2. Probability of correct response for high confidence R3 (green bar) and all ratings combined (brown bar) is shown for trials following high confidence responses (R3) only (**a**) and trials following low confidence response (R1 or R2) in (**b**). Due to the low number of R3 trials, the statistics in (**a**) is low, but a similar trend is seen in both graphs. There is no statistically significant difference between high confidence and all ratings combined probabilities of correct responses in (**a**) and (**b**).

3. We also investigated the effect of fatigue and habituation. As shown in the new Supplementary Fig. 5b the correct high confidence events appear to be evenly distributed throughout the entire session. This suggests that the involvement of some long-term effects have not played a significant role in our experiments. This fact was further supported by our observation (Fig. R1.3) that the distribution of correct responses for all ratings also remained constant throughout the session.

Figure R1.3. Distribution of ratings for correct post-selected single-photon events. All sessions were split into equal time intervals 0-20 minutes (blue), 20-40 minutes (green) and 40-60 minutes (red) following the first trial. The probability of correct response was obtained by averaging over all sessions and all subjects for the corresponding time intervals of the sessions.

As the reviewer states it would be interesting to address the role of other factors in future experiments in more detail, in particular those, which can occur within short timeframes, such as microsaccades, selective visual attention or phases of the brain's EEG. We have added the above points into the discussion of the revised manuscript (underlined changes on page 10).

Reviewer #2 (Remarks to the Author):

The authors present a novel light source that give them some control and knowledge over the number of photons delivered to the eye in a single flash. They use this light source to generate single photon stimulations to the eye. They find that a human's ability to detect a single photon stimulus is just a hair above chance, albeit significantly so.

The originality of this paper is in the light source, and the fact that it is used to address long-standing fundamental questions about human vision.

We thank the reviewer for her/his positive assessment and for acknowledging the originality of our work.

I do not necessarily doubt that it is possible for a human to sense a single photon. Extrapolation of psychometric functions from Hecht, Schlaer and Pirenne's 1942 paper suggests this possibility. However, the finding that the ability to detect a single photon is so slight, that even the slightest error or oversight in the analysis might lead to an erroneous result. Given that the authors are making such a profound statement, they owe it to the scientific community to be more convincing of this fact.

We fully agree with the reviewer that given the small size of the reported effect of single-photon detection, thorough experimental design, measurements and controls are required. We have tried to control for all conceivable alternatives to our experimental observations, as well as for systematic errors. Measures we have introduced include the implementation of the two-alternative forced-choice procedure (2AFC), which minimizes trial-to-trial variability by removing any effects caused by time-varying and subject-specific false-positive rates, elimination of any visual or auditory cues that subjects could receive during the experiment, frequent calibrations prior to each session, extensive training of subjects, as well as other measures, which are now more clearly highlighted and discussed in detail in the revised manuscript (see underlined changes in Methods and in Supplementary Note 4).

Moreover, as outlined in detail in the response to reviewer one as well as below, in the revised version of the manuscript we provide a set of additional experimental data and new analyses that corroborate our findings (see also our response to the first specific comment of reviewer one).

Finally, many questions that arise from read in the main text are only answered by reading the Supplemental Materials. This may not be easy to fix, except by publishing the paper in a journal that allows for all important information to be in the main text.

We fully agree with the reviewer and apologize for not being able to put all the important information into the main manuscript. In our revised manuscript, we have moved parts of the supplementary information that were key to following the text and for understanding our results into the main text (underlined changes on page 4 and 5 and in the Methods section).

While it's true that specialized journals often have less stringent space requirements, due to the fundamental and highly interdisciplinary nature of our study, we believe that Nature Communications represents the ideal platform for making our findings available to the broad community. We hope that the reviewer finds the balance between conciseness, detail and readability of our revised manuscript acceptable.

Specific Comments:

Page 4: Is it possible that the EMCCD detected a single photon when in fact there were two? Were the authors 100% certain when they selected the 'single photon trials'? It is not clear, even from the supplementary materials, that this is possible. According to Fig 1b, the detectors themselves record different numbers of photons, so one or both of the detectors are not 100% accurate. It is possible, therefore that some of the 'one photon' trials had more than one photon sent to the eye.

The reviewer raises an important point, which however was discussed in the original manuscript (Supplementary Note 3) but perhaps not with sufficient clarity. Naturally, there is a slight chance that in some of the 'one photon' trials more than one photon was sent towards the eye. In principle, this can only be excluded if the detection efficiency of the idler arm is 100%, which in reality is never feasible. We performed a careful

evaluation and comparison of the two most sensitive, commercially available, single-photon detector types, the single-photon avalanche diode (SPAD) and the EMCCD. As the most informative metric to compare the detectors we have chosen the ratio of single-to-multiple photon events, which is plotted in Fig 1b. As can be seen, the probability of multiple-photon events cannot be fully excluded, but the EMCCD outperforms the SPAD, since, after taking all factors into account, the SPDC source operated with the EMCCD leads to fewer undetected multiple photon events per one true single-photon event.

In addition, we also explicitly quantified the overall probability for the case asked by the reviewer, i.e. when one photon was detected by the camera *and* two or more photons were actually sent to the eye, reached the retina and induced two or more single-photon events in the retina. We find this probability to be negligibly low (0.02% - see revised manuscript pages 4, 5 and Supplementary Note 3) In our particular case, this probability was estimated from the efficiencies of the setup including multi-photon pair production rate, transmission and detection efficiencies. This suggests that out of all our 2420 post-selected single-photon trials on average less than one would have led to multiple signaling events at the retina. In the revised manuscript, we now highlight this in the main text (underlined changes on page 5).

Page 4 para 2: replace 'less' with 'fewer'

We thank the reviewer for noticing this error and we have corrected this in the revised manuscript.

Page 5, para 2: The authors report 2420 single photon events out of 30767 trials. This is 7.8 percent of trials, which is way more than the intended percentage of 4.8 reported in the previous page. Can the authors explain this?

Our source was indeed operated at a 4.8% mean photon production rate as stated in the text, i.e. out of 100 incoming UV pulses into the SPDC crystal on average 4.8 photon pairs were generated. The discrepancy between the chosen mean production rate of idler photons (4.8%) necessary to minimize generation of multi-photon events and the number of events passing the post-selection, i.e. 2420 out of 30767 (corresponding to 7.8%) is due to the intrinsic noise events of the EMCCD, which are indistinguishable from actual single-photon detections in the idler arm. We should note that this value as discussed in Supplementary Note 3 neither affects the actual pair production rate of 4.8% nor does it contribute to a higher probability of multi-photon events.

Page 5: The chance of reporting the interval containing the photon is just a hair above chance. The authors should be clearer in how they report this number. A reader might misinterpret the numbers reported as stating that subjects have just over a 50% chance of seeing a photon that strikes the eye, which is far from the case.

We thank the reviewer for drawing our attention to this potential source of misunderstanding by readers who may not be familiar with the nature of the two-alternative force choice (2AFC) procedure. In 2AFC even in the absence of any

information subjects answer correctly 50% of the time through random guessing alone. All reported probabilities of correct answers should therefore be compared relative to the 50% random guessing baseline. We have followed the reviewer's suggestion and indicated this clearly in the main text, revised figures and figure captions.

Page 9, last para: It seems very unlikely that EEG or MEG will contribute anything to understanding this this phenomena. It's recommended to remove this section.

We agree with the reviewer that we have not sufficiently explained how EEG or MEG might contribute to the understanding of the phenomena. While it is unlikely that such measurements will result in a fundamental understanding of the mechanism underlying single photon vision, one possibility, as suggested by reviewer 1, may be that parallel recordings via EEG and MEG may allow for the identification of whether a correct response correlates with other top-down processes, such as visual attention or phases of the brain oscillations etc. We have followed the reviewer's suggestion and toned down the wording to hint at such experiments as a potentially interesting direction for future studies (underlined changes on page 10).

Page 11, line 1. "...barely visible red light...". The control of the light levels is critical for this experiment, so it is surprising that the authors provide no details, or express no concern over the light used for fixation. What is the wavelength of the fixation light? What is the power? What is the bandwidth? Given typical levels of light scattering in a human eye, what number of photons from this source will land on the tested area that is 23 degrees away? Given the spectral sensitivity of the rods, what is the probability that these photons might be seen?

We thank the reviewer for highlighting the lack of this important information in the original manuscript. The fixation light was on during the initial alignment of each session. Later during the session subjects could switch it on whenever they wanted to make sure their alignment was correct or to check if they felt they may be losing it.

The fixation light was directed to the central fovea and the fraction of light which scatters from here into the region of the retina where our SPDC light was focused (23° temporal), can be estimated by using the point spread function (PSF) of the human eye. The scatter specific part of the PSF in the human eye for angles ranging from 3 to about 30 degrees is approximated by the formula $P(a)=S/a^2$, where a is the angular distance from the central part of the PSF and S is the fit parameter, which has been measured experimentally [Ginis et al., Biomed Opt Express, 2014, 5:3036-41; Westheimer and Liang, J Opt Soc Am A Opt Image Sci Vis. 1995, 12:1417-24]. Given the optics and geometry used in our experiment (see Methods) this results in ~ 0.0002 of the fixation light power being scattered into the test area.

The wavelength of our fixation light was 660 nm with a FWHM of 5 nm and it was operated at an output power of $\sim 6 \mu\text{W}$ output power. The power was further reduced by a set of ND filters with overall OD 6.5 and directed at the central fovea. This led to ~ 3.2 million photons entering the eye during a 1 s time-period. Therefore, during the rod integration time of 0.1 s approximately 60 photons were scattered onto the retinal patch onto which also the single photon beam was impinging. Effect of these photons in

contributing to a photo-isomerization event is further reduced by the very low spectral cross section of rod cells at long wavelengths which at 660nm is only $\sim 0.000003\%$ of its maximum at 500nm [CIE Proceedings, 1951, Vol. 1, Sec 4; Vol 3, p. 37]. Therefore, the effect of these 60 photons impinging on our considered retinal patch containing predominantly rod cells could be compared to the effect of ~ 0.02 additional experimental SPDC photons. Moreover given the extreme low density and low sensitivity of cones in the considered retinal area the effect of these photons on cones can be ignored. While as mentioned above the fixation light was only occasionally on, this estimate shows that we can safely neglect interference with our results from the fixation light, even if it were to have been on continuously.

We have updated the Methods section of the revised manuscript (underlined changes on page 11 in the Methods section) and Supplementary Note 4 to contain this information.

Page 11: para 2, line 1: Who triggered each trial? Was is the subject (ie self-paced)? What was the average time taken for each trial.

Again, this important information was indeed missing in our original manuscript. The subjects triggered the trials by themselves and were instructed to proceed at whatever pace they felt comfortable with and to take regular breaks if and whenever desired. The average time between the trials in our experiments was 2.5 s. At some trials subjects paused for long periods and were allowed to stop the session entirely if they felt tired or wanted to do so.

This information has been added into the Methods section of the revised manuscript (underlined changes on page 12).

Page 11: para 2, last line: Why was feedback used? This is a crucial issue here. It makes sense that feedback was used for training (Suppl. Mat), but not for the experiment. With feedback, it is possible that the subjects could have used other cues, however subtle, to know when a photon was emitted. Did the experimenters try the experiment with no feedback? If yes, then what were the results? If not, then they should.

The reviewer has raised an important point. Feedback is an essential part of the training as it helps subject to maximize the performance under extreme light conditions as shown in Supplementary Fig. 4. However, we also found that feedback during actual experimental trials is useful as it helped subjects to stay motivated and alert throughout the entire session.

There is a known trade-off in the visual system between sensitivity and reliability. The sensitivity can be increased by asking subjects to use more lenient criteria and to not be afraid of making mistakes [Sakitt, B. 1972. J Physiol 223, 20; Teich, et al., 1982. J Opt Soc Am 72, 1]. In our experiments subjects received in $\sim 50\%$ of trials (due to the nature of the 2-AFC protocol) a positive feedback. This helped keeping subjects motivated and focused on the task while the use of our rating scheme allowed to push subjects to their maximum sensitivity [Sakitt, B. 1972. J Physiol 223, 20].

We agree with the reviewer that a prerequisite is however to ensure that subjects did not use any other cues that allowed them to know when a photon was emitted. As outlined in

detail in the revised Methods section (underlined changes on page 11), we took great care in designing the experiment such that no such putative cues, visual, auditory or otherwise could have been available to the subject. Below we provide a brief description of some of these measures and why we believe that the presence of such cues can be excluded in our experiments.

Subjects were seated in the lightproof box in which the design did not allow any light except the stimulus pulses and fixation light to enter. The subjects wore headphones through which they could only hear the beeps that heralded the light and control pulses respectively as described in detail in Methods section. In addition, the ~2 cm synthetic foam-filled panels of the box provided good acoustic isolation. Moreover, the room in which the box was located was dark and quiet. Given this acoustic and visual isolation of the subjects, the fact that all control elements of the experiment did not produce any sound and that there were no other sensorially perceivable asymmetries between the first and second time interval of our 2AFC protocol than the light stimulus itself, it is difficult to see how subjects may have used alternative cues to gain information on which interval could have contained the stimulus. On the contrary, exactly because most of the time subjects did not see or hear anything without any positive encouragement as provided by the feedback, an experimental session lasting up to 2.5 hours would become extremely daunting and tiring, negatively affecting the performance of the subjects.

Furthermore, we could directly show the absence of any cues by analyzing the performance of subjects in trials where no photons were detected by the EMCCD. In the majority of these trials (~95%) no photon pair was generated and no photons were sent to the subjects' eye. If cues other than the light itself contributed to identifying the stimulus time interval, the subjects' performance in these trials containing no real photons would have been comparable to the performance in our post-selected single-photon trials. However, the probability of correct response for these blank trials is not different from the 0.5 baseline for both combined and high confidence ratings (0.505 ± 0.003 , p -value = 0.08 and 0.507 ± 0.01 , p -value = 0.3, respectively). This is what would be expected for random guessing in the absence of any cues (see underlined changes in Supplementary Note 4).

Nevertheless, we have also tried to answer the issue concerning feedback as raised by the reviewer more directly and through additional experiments. However, experiments with the single-photon quantum light source are extremely demanding and were collected over a time of several months. Therefore, additional experiments with our SPDC source without feedback would not have been feasible within the available timeframe for a revision.

Instead, we have used a classical Poissonian light source which allowed for a significantly larger number of trials to be obtained within a feasible time frame for a revision. We collected a total of 905 additional trials without feedback and 1875 trials with feedback at an average of one photon incident at the cornea. In both cases, the data showed a similar statistical trend as in the SPDC data and the subject's performance did not depend on the presence of feedback (Fig. R2.1. below). We therefore conclude that while feedback helped subjects to stay focused on the task and kept them motivated, it did not have a significant detectable effect on the subjects' performance.

Figure R2.1. Probability of correct response in experiments with a Poisson light source emitting an average of one photon per pulse with feedback ((a), n=1875) and without feedback ((b), n=905).

It is nonetheless in principle possible that positive and negative feedback could differentially influence the performance of the subjects in the subsequent trial through cognitive effects. In order to test for such possibility we analyzed events in which subjects received negative or positive feedback in the previous trial separately. We found no significant correlation between the given feedback (positive or negative) and the probability of correct response or the distribution of the confidence ratings (Supplementary Fig. 6). We therefore concluded that within our experimental setup there is no unspecific or general cognitive phenomenon influencing the performance of the subjects that would differentially depend on the received feedback.

Page 11, para 4: 1000 trials over two hours is 7 seconds per trial. If the 2 hours includes the ~ 1 hour of dark adaptation, then the average time per trial was 3.6 seconds. Given the long time between trials, how can the authors get a data point for probability of a correct response at 2 seconds between the current and preceding event (Fig 2(c&d))?

This information was indeed missing in our original manuscript. The average time between the trials in our experiments was 2.5 seconds. However, the distribution was quite broad (standard deviation of 0.6 s), since the pace of the experiments was controlled by the subjects. The time between individual trials was below 2 seconds for ~ 10% of all trials. In a small number of trials (less than 3%) subjects paused for long periods, which increased the overall duration of the session, leading to an overestimate for the average time between trials when simply dividing the length of the session by the number of trials, as done by the reviewer. In the revised version of the manuscript we have added this information to the Methods section and have corrected the text accordingly.

Page 12, bottom: Contrary to what is written in the methods, it appears from figure 1 that there is no fiber in the idler arm but that there is a single mode fiber in the Signal arm.

What are the coupling losses of this fiber? My understanding is that they are about 75% at best. It seems that there may be more events where a photon was detected by the EMCCD but its paired photon never makes it to the eye.

We apologize for the typo that appears in the Methods section. As drawn in Fig. 1a, there is a single mode fiber in the signal arm of our setup (to guide the photon to the eye) but there is no fiber in the idler arm. We have corrected this typo in the revised Methods section (underlined changes on page 14).

Regarding the reviewer's comments on coupling losses, the coupling losses from a SPDC source into a fiber are indeed rather high, and we estimate them to be ~60%. When combined with the detection efficiency of the EMCCD (~90%) and its noise (clock-induced charge rate ~0.04) this is what we have referred to as the "heralding efficiency" (i.e. the probability for a single photon impinging on the cornea provided its partner is detected by the EMCCD) and which is discussed in detail in the original Supplementary Note 1 and 3. The heralding efficiency in our setup is ~20%, so indeed there are many events 'where a photon was detected by the EMCCD but its paired photon never makes it to the eye'. This fact was already taken into account in the original manuscript and does not influence the main conclusion of our work, i.e. the ability of subjects to detect single photons and the validity of our observation on single-photon induced modulation of the sensitivity of the visual system. This is because a low heralding efficiency would only lead to a higher number of blank vs. blank trials for the subjects but would never lead to *more* than a single photon at the subjects' cornea. This is discussed in detail in Supplementary Note 3.

Page 13, para 2: Can the authors be certain that '..one and only photon..' events are determined accurately by the EMCCD? Is it possible that a multiple photon even would be mis-identified as a single photon event?

The reviewer raises an important point, which we also discuss further above and in the revised Supplementary Note 3. In order to ensure a high fidelity of single photons delivered to the eye it was important to quantify and minimize the frequency of the multiple photon events that could reach the eye and lead to the multiple retinal signals in post-selected trials that were identified by EMCCD as single-photon events.

One of the key advantages of using an EMCCD to detect the single-photon SPDC events, was that its multi-pixel design allowed for the identification of cases in which two or higher number photon states were generated. These events could then be excluded from further analysis through post-selection. Given the high detection efficiency of the EMCCD and its 2D detector architecture the majority of such two and multi-pair events could be successfully identified. While at our optimally chosen photon-pair production rate of 0.048 (see Main text and Supplementary note 3) the rate of two and higher-pair events in the signal arm was already highly reduced (~0.11%), using this scheme allowed us to detect ~80% of such cases where two or a higher number of photons arrived at the EMCCD (Supplementary Note 1 and 3).

This ability of the EMCCD to identify multi-photon events, together with the efficiency of the signal arm, the transmission efficiency of the ocular medium and the quantum efficiency of the photo isomerization left us with only ~0.02 % of post-selected trials in which two or more photon states were generated by the crystal, misidentified by the EMCCD camera as a single photon event and elicited a two (or more)-photon signal on the retina (see also our response on the first specific comment). This means that from our 2,420 single-photon trials out of a total ~30,000 that passed the post selection on average only for less than one trial would have got such an incorrect assignment. We thus feel that this effect can be safely ignored.

Figure 2: What are the error bars?

We apologize for not being sufficiently clear on this point. Unless stated otherwise all error bars are standard error of the mean (SEM). We have made this clear both in the figures and in the revised text.

Supplementary Table 1: Please specify the wavelengths that these numbers refer to.

We agree with the reviewer that this is important information and we added it to the revised Supplementary Table 1.

Reviewer #3 (Remarks to the Author):

While the sensitivity of human eyes to light has been determined to be on the few photon level a few decades ago in a number of celebrated experiments. The absolute lower limit was not known. The use of a pair of correlated photons, the authors develop a novel experimental scheme that they can ensure that they can identify trials where only a single photon was incident up the retina with unprecedented accuracy (approximately an order of magnitude better than traditional Poisson noise). This is a very novel application of a correlated photon source. The exciting result is that our eyes is responsive to single photon with probability slightly higher than random. It is also interesting that they find a "potentiation" effect where detection of one photon increase the probability of future detection within some time constant. Overall, the experiments are well designed with appropriate simulation and theoretical estimate to bolster the experimental results. The paper is well written and clearly presents these original findings.

We thank the reviewer for the very positive assessment of our work.

I have only few minor comments:

1. While it is convincing that R3 responses has a probability of 0.59 ± 0.04 , the combined responses has only a probability of 0.51 ± 0.01 . Unless I am missing something, it does appear that the combined probability is above random chance.

The reviewer is probably referring to the Fig. 2c, which shows that single photon detection is enhanced by the previous photon, and the low performance cited by the reviewer suggest that she/he refers to the events with long delays (over 10 s). We fully agree with the reviewer that for these events the performance is not significantly above random chance and we have modified the text to make this clear (underlined changes on page 7). The probability of correct response is higher and the corresponding performance is significant for shorter delays (Fig 2c, p value = 0.02, for combined data at < 10 s for combined ratings) and in agreement with the reviewers comment it is significant for high confidence data irrespective of the delay (Fig. 2c inset). We have modified the text to state this clearly (underlined changes on page 7).

2. Please explain why if read noise of EMCCD is 200 count and 220 count corresponds to 6 sigma. What is the underlying statistic of this 200 count read noise (it is not Poisson for sure. Does the 200 count include some pedestal value?)

We thank the reviewer for pointing out issue, which was poorly described in our original manuscript. The value of 200 counts is indeed an arbitrary offset value, which is inbuilt to the camera as a result of the manufacturing process, and about which the counts are normally distributed. We empirically determined the standard deviation of this distribution to be ~ 3.5 counts (distribution shown in Fig. R3.1) by operating the camera in the absence of stimulus light in a dark chamber and with its external shutter and lid closed. This offset and normal broadening of the count values occur at the final stage of the EMCCD read-out and is intrinsic to the performance of the camera and did not affect the quality of our source as a 6-sigma rule was applied with respect to this pedestal value. In the revised manuscript, we have modified the Supplementary Note 1 to make these points more clear.

Figure R3.1. Measured distribution of EMCCD counts in the absence of light and with the camera shutter and lid closed. Both experimental data (blue) and theoretical simulation (red) are shown.

3. The use of EMCCD as a large number of parallel detector to ensure multiphoton events can be distinguish from single photon event is an important innovation in the

experiment. It is expected that it should be improved upon single pixel detector like SPAD but does this improvement dependent on the correlated photon generation geometry and the intermediate optics between the photon source and the EMCCD?

The reviewer is indeed correct that the improvement the EMCCD offers over SPADs depends on the experimental configuration. As mentioned in Supplementary Note 1, the dominant source of noise source in our experiment is the clock-induced charge of the EMCCD, the rate of which increases linearly with pixel number. Using too many pixels would quickly swamp the signal in this noise. It is therefore desirable to minimize the number of pixels used, but this of course decreases the ability to discriminate between single and multi-photon events by increasing the probability of multiple photons landing on the same pixel. We therefore selected, following experimental tests and numerical simulations detailed in Supplementary Note 2, a configuration where the idler light was focused down with a short focal length lens to cover a 3x3 pixel array. In this configuration the EMCCD could detect multiple photon events in 80% of cases (Supplementary Note 3), while the probability of the EMCCD registering a single photon while two photons produce a signal in the retina is only $\sim 0.02\%$ and could thus be safely ignored (see Supplementary Note 3, also response to the specific comment 1 of the reviewer 2)

With regards to the photon generation geometry, the experiments relied upon the spatially non-degenerate production of the signal and the idler beams, but this is not unique to using an EMCCD and would be required regardless of the detection mechanism.

REVIEWERS' COMMENTS:

Reviewer #1 (Remarks to the Author):

I'm impressed with the additional control experiments and with the detailed, thoughtful and adequate replies to all of my comments and queries. Their experiment is a beautiful one and ought to be published.

Christof Koch

Reviewer #2 (Remarks to the Author):

Thanks to the authors for clarifying my questions on this paper. The authors have satisfactorily addressed my questions.

Specifically, my concern about the use of feedback has been addressed, albeit not directly. I maintain that the use of feedback as a motivational tool is not acceptable and will not be acceptable to many readers. However, the fact that feedback was used on ALL trials (not clear in the original manuscript) does actually strengthen the paper, even though the authors may not have intended it. Let me explain.....

In the main methods (p12) the authors write "Subjects received feedback as to whether or not their response was correct." This is confusing, since most trials contain no photons and so there is no actual 'correct' response for a vast majority of the trials. However, the non-photon trials effectively yield what are called 'catch' trials, where any bias in the response of the subject can be evaluated. It is the chance selection of the 'catch' trials that seals the deal for me. It would clarify the paper if the authors stated that the feedback was given as to whether the subject chose the interval that was randomly selected to contain the stimulus, WHETHER THE STIMULUS WAS PRESENT OR NOT. I also urge the authors to move the statistics of the subject's responses to 'catch' trials to the main section of the paper. Providing that detail in the main body of the text will save the skeptical reader a lot of time.

REVIEWERS' COMMENTS:

Reviewer #1 (Remarks to the Author):

I'm impressed with the additional control experiments and with the detailed, thoughtful and adequate replies to all of my comments and queries. Their experiment is a beautiful one and ought to be published.

Christof Koch

We would like to thank Prof. Koch for this positive assessment of our work and for his very constructive and helpful comment during the review process.

Reviewer #2 (Remarks to the Author):

Thanks to the authors for clarifying my questions on this paper. The authors have satisfactorily addressed my questions.

Specifically, my concern about the use of feedback has been addressed, albeit not directly. I maintain that the use of feedback as a motivational tool is not acceptable and will not be acceptable to many readers. However, the fact that feedback was used on ALL trials (not clear in the original manuscript) does actually strengthen the paper, even though the authors may not have intended it. Let me explain.....

In the main methods (p12) the authors write "Subjects received feedback as to whether or not their response was correct." This is confusing, since most trials contain no photons and so there is no actual 'correct' response for a vast majority of the trials. However, the non-photon trials effectively yield what are called 'catch' trials, where any bias in the response of the subject can be evaluated. It is the chance selection of the 'catch' trials that seals the deal for me. It would clarify the paper if the authors stated that the feedback was given as to whether the subject chose the interval that was randomly selected to contain the stimulus, WHETHER THE STIMULUS WAS PRESENT OR NOT. I also urge the authors to move the statistics of the subject's responses to 'catch' trials to the main section of the paper. Providing that detail in the main body of the text will save the skeptical reader a lot of time.

We thank the reviewer for the careful explanation. We agree that chance selection of the trials, in which subjects were not presented a photon but still received a feedback, is essential to evaluate response biases of the subjects. Following the reviewer suggestion, we added this information to the main text and the Methods section of the revised manuscript.